# VIDEO ACTION DIFFERENCING

**James Burgess**[1], **Xiaohan Wang**[1], **Yuhui Zhang**[1], **Anita Rau**[1], **Alejandro Lozano**[1],
**Lisa Dunlap**[2], **Trevor Darrell**[2], **Serena Yeung-Levy**[1]
[1]Stanford, [2] UC Berkeley

## ABSTRACT

How do two individuals differ when performing the same action? In this work, we introduce Video Action Differencing (VidDiff), the novel task of identifying subtle differences between videos of the same action, which has many applications, such as coaching and skill learning. To enable development on this new task, we first create VidDiffBench, a benchmark dataset containing 549 video pairs, with human annotations of 4,469 fine-grained action differences and 2,075 localization timestamps indicating where these differences occur. Our experiments demonstrate that VidDiffBench poses a significant challenge for state-of-the-art large multimodal models (LMMs), such as GPT-4o and Qwen2-VL. By analyzing failure cases of LMMs on VidDiffBench, we highlight two key challenges for this task: localizing relevant sub-actions over two videos and fine-grained frame comparison. To overcome these, we propose the VidDiff method, an agentic workflow that breaks the task into three stages: action difference proposal, keyframe localization, and frame differencing, each stage utilizing specialized foundation models. To encourage future research in this new task, we release the benchmark[1] and code[2].

## 1 INTRODUCTION

The ability to compare two videos of the same action and discern their detailed differences plays a critical role in a wide variety of applications. For instance, in fitness coaching, a novice learning to perform a barbell squat typically watches instructional videos and then compares their actions in a recorded video to identify discrepancies between their movements and those of an expert. In medical training, junior surgeons compare videos of themselves performing surgical procedures with reference videos from experts to identify errors and improve surgical skills.

There are two critical obstacles. First is precise *localization of sub-actions*: finding differences requires finding the sub-action frames where the differences might occur, and aligning those frames between the two videos. Second is *fine-grained understanding*: the ability to perceive subtle visual differences in motions.

Current research on video difference understanding largely emphasizes feature visualization (Balakrishnan et al., 2015) or coarse-grained comparisons between different actions or interacting objects (Nagarajan & Torresani, 2024). However, many real-world applications demand fine-grained comparisons between videos of the same action, a challenge that has received little attention.

We introduce a new task, Video Action Differencing (VidDiff). Given two videos of the same action, $(v_A, v_B)$, along with a description of the action, the task is to generate two sets of statements: one that is more true for $v_A$ and another for $v_B$. For example, in a video pair featuring an expert and a novice performing a barbell squat, key differences might include "knees caving in more in video A" and "the squat is deeper in video B" (Figure 1). Since generating the initial difference candidates relies heavily on language capabilities, we also introduce a simpler 'closed' setting that focuses on video analysis. In this setting, the target difference strings are provided, and the task is to predict whether each applies more to video A or B.

To facilitate research in this new direction, we present VidDiffBench, a comprehensive benchmark designed for video action differencing. VidDiffBench contains 549 video pairs drawn from domains

---

[1]Benchmark: https://huggingface.co/datasets/jmhb/VidDiffBench
[2]Project page: http://jmhb0.github.io/viddiff

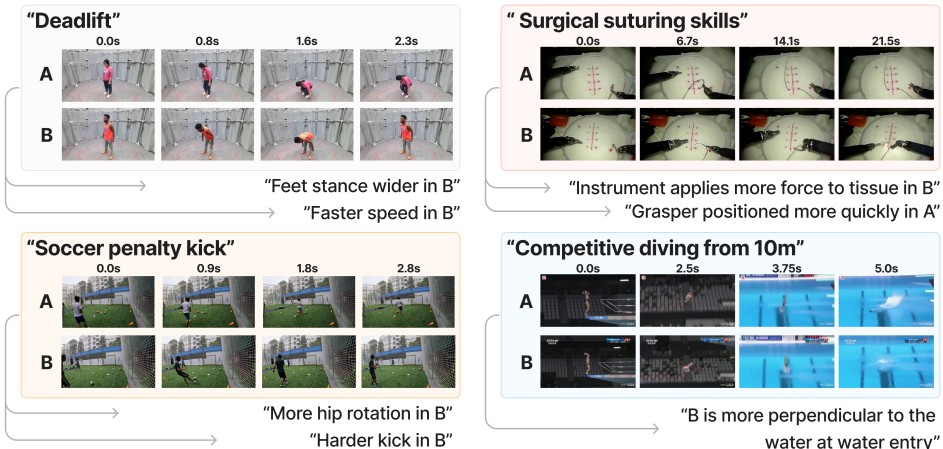

Figure 1: The Video Action Differencing task and benchmark (VidDiffBench). Given a pair of videos and an action, the task is to generate a list of differences as natural language descriptions. Our VidDiffBench consists of annotated differences across diverse domains, where the differences are relevant to human skill learning. The first row emphasizes the first key challenge: *localization of sub-actions* between segments of the video for comparison. The second row highlights the second key challenge: *fine-grained image understanding* of actions in order to perform comparison.

that require expert feedback, such as fitness, sports, music, and surgery. The videos are annotated with 4,469 fine-grained differences (∼8 per video pair), along with 2,075 timestamp annotations that identify where these differences occur. To ensure the annotated differences are relevant to skill learning, we create a taxonomy of action differences that leverages domain expertise. This makes VidDiffBench the first large-scale dataset dedicated to video action differencing.

In addition to introducing a new task and benchmark, we propose the VidDiff Method, an agentic workflow (Anthropic, 2025) for solving VidDiff in zero-shot. The method incorporates large language models (LLMs) to propose differences, localizes relevant frames using contrastive language-image models (CLIP), and compares frames for differences using vision-language models (VLMs). The key idea is that localizing specific video segments where differences occur enables more effective visual comparison with VLMs. We further benchmark both open-source (Qwen2-VL, LLaVA-Video) and proprietary (GPT-4o, Gemini-1.5 pro, Claude 3.5 Sonnet) large multimodal models (LMMs) on VidDiffBench. Our results demonstrate that VidDiff is a very challenging task for zero shot models, while the structured approach in the VidDiff Method enhances video comparison.

## 2 RELATED WORK

**Skilled Action Understanding in Videos** Video comparison has many potential applications, and our benchmark focuses on the specific goal of natural language feedback in skill learning. Most of the video action comparison papers from this section's first paragraph are systems for skill feedback, showing that skill feedback is well-motivated. Many works give feedback by classifying coarse motion errors, or by visualizing motions, with applications in yoga (Zhao et al., 2022; Thoutam et al., 2022; Chen et al., 2018; Dittakavi et al., 2022; Chen & Yang, 2020; Xie et al., 2019), physical therapy (Velloso et al., 2013), weightlifting (Parmar et al., 2022; Ogata et al., 2019), and general fitness (Fieraru et al., 2021; Ashwin et al., 2023). The feedback tends to be coarse-grained. In contrast, our task focuses on open natural language feedback, and identifying fine-grained feedback. Recently, the Ego-Exo4D dataset (Grauman et al., 2023) provides videos with expert commentary on skilled actions, which is promising for developing instructional feedback systems. This, along with existing works that give language feedback (Li et al., 2024b; Fieraru et al., 2021; Parmar et al., 2022; Velloso et al., 2013), support our claim that language is a good medium for providing skill feed-

back to humans. Zooming out from skills feedback, skilled action understanding – which includes foundational capabilities for feedback systems – has attracted enormous interest. For example, in sports, music, dance, and surgery, prior works have tackled action recognition (Verma et al., 2020; Shahroudy et al., 2016; Soomro et al., 2012; Zhang et al., 2013; Wang & Zemel, 2016; Chung et al., 2021); spatial and temporal action localization / segmentation (Shao et al., 2020; Liu et al., 2022; Li et al., 2021b; Zhang et al., 2023b; Ibrahim et al., 2016; Garrow et al., 2021; Li et al., 2021b; Aklilu et al., 2024); human pose and motion estimation / reconstruction (Cai et al., 2022; Tang et al., 2023b; Wang et al., 2023; Andriluka et al., 2014; Li et al., 2021a; Fieraru et al., 2021; Zhu et al., 2022; Bera et al., 2023; Liu et al., 2024; Grauman et al., 2023); and hand and tool pose estimation (Doosti, 2019; Johnson et al., 2020; 2016; Gao et al., 2014; Grauman et al., 2023). Skilled action domains also tackle higher level reasoning tasks like question answering (Li et al., 2024a), and action quality assessment (Pirsiavash et al., 2014; Parmar & Tran Morris, 2017).

**Visual Difference Understanding**   Only a few prior works have considered video comparison in actions. They mostly emphasize skill learning in similar categories to our benchmark, but their methods tend to tackle single domains. One approach visualizes the user's motion against a target expert motion in video or in augmented reality (AR) (Trout, 2013; Motokawa & Saito, 2006; Han et al., 2016; Kyan et al., 2015; Kurillo et al., 2008). Since interpreting discrepancies between motions is challenging, especially for novices, other works generate visualizations of differences (Liu et al., 2023; Liao et al., 2023; Balakrishnan et al., 2015). In contrast, we summarize action differences in natural language, which enables direct and interpretable feedback. Also, our benchmark covers many skill categories, encouraging the development of generalizable methods that do not require domain-specific training data and methods. The most related work by Nagarajan & Torresani (2024) focuses on coarse-grained step differences in instructional videos using question-answer pairs. In contrast, our approach targets fine-grained action differences, such as a "deeper squat", which offers more detailed insights for skill learning. Additionally, our VidDiff method is zero-shot for a benchmark spanning multiple skilled domains, while their method requires instruction tuning data and is specialized to cooking. Beyond inference-time comparison, a number of important works in skill assessment leverage video pairs in training – here the supervision signal is typically a binary variable indicating which video demonstrates greater skill Doughty et al. (2018; 2019); Pan et al. (2021); Zhang et al. (2023a). In appendix E, we discuss all related datasets having video pairs, finding that none have labels for fine-grained comparison while being large scale, unlike our VidDiffBench

Describing differences between *images* in language is an established task called 'difference captioning' or 'change captioning' (Jhamtani & Berg-Kirkpatrick, 2018; Park et al., 2019; Kim et al., 2021; Yao et al., 2022; Hu et al., 2023). LMM evaluation and instruct-tuning papers address image differencing for pairs or small sets of images (Alayrac et al., 2022; Li et al., 2023; Achiam et al., 2023; Jiang et al., 2024). The task of image set differencing with large sets was introduced in (Dunlap et al., 2023). Our VidDiff method uses image differencing with LMMs as a subroutine, however the task of video action differencing with natural language has not previously been explored.

## 3 VIDEO ACTION DIFFERENCING

Video Action Differencing (VidDiff) is a novel and challenging task, offering significant potential for applications in coaching, skill acquisition, and automated performance feedback. To facilitate the development of models capable of handling such a task, we define two complementary task settings: a *closed* setting, evaluated via multiple-choice format, and a more complex *open* setting, requiring generation of action differences. Both are essential for advancing video understanding, especially in contexts where precise feedback on actions is critical.

### 3.1 TASK DEFINITION

The goal of video action differencing is to identify skill-based differences between two videos of the same action, in a zero-shot setting. We first introduce the simpler *closed-set* version, followed by the more difficult *open-set* variation.

**Closed-Set Video Action Differencing:** In the closed-set task, the input is an action description string $s$, a video pair $(v_A, v_B)$, and a list of $k$ candidate difference statements $\mathbf{D} =$

$\{d_0, d_1, \ldots, d_{k-1}\}$, such as "the jump is higher". For each $k$, the model makes a predictions $\mathbf{P} = \{p_0, p_1, \ldots, p_{k-1}\}$, where each prediction is either 'A' if the statement applies more to $v_A$, or 'B' if it applies more to $v_B$. This setup simulates real-world scenarios like coaching, where specific differences of interest are already known. For benchmark purposes, the dataset only includes instances where there is a clear ground-truth label ('A' or 'B') for each difference, which makes evaluation both reliable and automatic.

**Open-Set Video Action Differencing:** In the open-set task, the input is the action description string $s$, a video pair $(v_A, v_B)$, and an integer $N_{\text{diff}}$. The model must generate at most $N_{\text{diff}}$ difference statements $\mathbf{D}$ and their associated predictions $\mathbf{P}$, which label the differences as 'A' for video $v_A$ or 'B' for video $v_B$. This setting is more challenging, as the model must not only identify differences, but also generate those differences without any pre-defined options, closely mimicking real-world conditions.

## 3.2 Evaluation Metric

Our choice of benchmark evaluation metrics is driven by two major challenges for designing annotations: *ambiguity* and *calibration*. First, there is ambiguity around what differences are important for performing an action skillfully. Second, annotators are calibrated differently – they have different thresholds for whether a difference like "wider feet stance" is different enough to be annotated.

**Closed-Set Evaluation:** In the closed-set task, the evaluation is straightforward: prediction accuracy is measured as the percentage of correct predictions, where 50% corresponds to random guessing and 100% represents perfect performance (assuming a balanced evaluation set). There is no *ambiguity* because we provide the possible differences. There is no *calibration* issue because the answer must be 'A' or 'B' (and not 'C' for "not different"). Overall, it's an automatic metric that focuses on video understanding.

**Open-Set Evaluation:** In the open-set task, we use an LLM query (GPT-4o) to match the ground truth difference strings to predicted difference strings in a 'partial matching'. Then we only consider "positive differences" – where the ground-truth label is 'A' or 'B' and not 'C'. Then the recall@$N_{\text{diff}}$ is calculated as the number of correctly matched and predicted positive differences, divided by the total number of positive differences. To handle the *ambiguity* of what differences are relevant, we set $N_{\text{diff}}$ to be 1.5 times the number of labeled differences, so models can predict more differences without penalty. This is a reasonable number because the annotation taxonomy is designed to cover skill-relevant differences. Moreover, we handle the *calibration* challenge of whether a difference is 'above a threshold' by only considering the positive differences where ground truth is 'A' or 'B'.

## 4 Benchmark Dataset and Annotations

The Video Action Differencing task presents a novel challenge in video understanding, requiring precise comparison of subtle action differences. As no comprehensive benchmark to evaluate this task exists, we introduce VidDiffBench, a comprehensive benchmark specifically designed to test and advance the ability of models to detect fine-grained differences in complex actions. Our benchmark consists of publicly available videos and our human-created annotations are freely available on HuggingFace Hub[3]. VidDiffBench covers a wide range of actions relevant to skill learning and performance feedback, and is constructed to challenge models across varying levels of difficulty, ensuring its relevance for long-term model development. Table 4 summarizes the key dataset statistics.

### 4.1 Video Datasets

The video collection for VidDiffBench was designed to capture a diverse range of actions where performance feedback is essential, ranging from simple exercises to complex professional tasks. This diversity ensures that models are challenged not only on temporal localization but also on the subtlety and complexity of visual differences. Actions in VidDiffBench span multiple levels of difficulty—from the basic "hip rotations" in fitness exercises to the intricate "surgical knot tying." This wide coverage tests models across varying degrees of granularity and action complexity. The are five categories:

---

[3]https://huggingface.co/datasets/jmhb/VidDiffBench

| Category | Source Dataset | Activity | Video Pair | Difference | Timestamp |
|---|---|---|---|---|---|
| Fitness | HuMMan (Cai et al., 2022) | 8 | 193 | 1,466 | 310 |
| Ballsports | Ego-Exo4d (Grauman et al., 2023) | 4 | 98 | 996 | 595 |
| Surgery | JIGSAWS (Gao et al., 2014) | 3 | 166 | 1,386 | 672 |
| Music | Ego-Exo4d (Grauman et al., 2023) | 2 | 29 | 180 | 320 |
| Diving | FineDiving (Xu et al., 2022) | 1 | 63 | 441 | 140 |
| Total | | **18** | **549** | **4,469** | **2,075** |

Table 1: Summary of VidDiffBench statistics across categories and datasets: number of unique activities, video pairs, annotations for differences, and timestamps.

- *Fitness* videos are simple, single-human exercises sourced from HuMMan (Cai et al., 2022), characterized by clean consistent backgrounds, consistent camera viewing angles, and consistent movement patterns.

- *Ballsports* includes basketball and soccer actions from Ego-Exo4D (Grauman et al., 2023), recorded across various environments with some diversity in background and camera angle, as well as action detail.

- *Diving* features high-level Olympic performances from the FineDiving dataset (Xu et al., 2022), capturing subtle and complex movements in professional diving. The backgrounds may be different, but the camera angles are consistent.

- *Music* contains guitar and piano exercises from Ego-Exo4D (Grauman et al., 2023), focusing on detailed finger and hand movements. Background and camera angles can vary.

- *Surgery* includes long, intricate procedures such as "knot tying" and "needle passing" from the JIGSAWS dataset (Gao et al., 2014). The background and camera angles are consistent.

Within each action, video pairs are randomly sampled to ensure a wide range of comparison difficulty. The range of tasks is broad in terms of action complexity and background variation.

## 4.2 VIDEO ACTION DIFFERENCE ANNOTATIONS

A critical innovation of VidDiffBench is its detailed human-annotated dataset, designed to address two major challenges in annotating the video differencing task: *ambiguity* in identifying relevant differences and *calibration* consistency among annotators. To tackle ambiguity, we introduce a structured difference taxonomy for each action, ensuring clarity on what aspects are being compared. Then we assign annotators to label video pairs with differences – to handle the calibration challenge we ensure labeling consistency by maintaining a consistent annotator identity within each action. Additionally, we provide frame-level localization annotations of differences, which can enable analysis for future model development. In the following section, we describe these components in greater detail.

### 4.2.1 ANNOTATION TAXONOMY

For each action, we define a structured *difference taxonomy* – a list of key visual differences relevant to the task. For instance, in the basketball jump shot, a skill-relevant difference might be "the ball is more in front of the body"; on the other hand, we do not include differences not directly relevant to skill performance like "the athlete is taller". Annotators assign labels to video pairs as follows: 'A' if the difference is more pronounced in video A, 'B' if it's more pronounced in video B, and 'C' if the difference is negligible. By fixing this taxonomy, we address the *ambiguity* challenge – that different annotators may not focus on the same differences. This allows for more objective and consistent comparisons.

We consulted domain experts to create the taxonomies for each action category. For Fitness and Surgery, we worked with a personal trainer and an attending surgeon, respectively, to identify visually salient differences between novice and expert performers. For Ballsports and Music, we extracted relevant differences from expert commentary in the Ego-Exo4D dataset using a large language model (LLM). For Diving, we leveraged the FINA diving manual, processed by an LLM, to

identify key differences. We filtered differences that were difficult to visually assess, such as "more wrist snap" in basketball jump shot (because video resolution was not high enough).

This method resulted in 148 distinct difference descriptions, which are detailed in Appendix G.2. This fixed taxonomy allows for precise evaluation of model performance across video pairs and helps identify failure cases where models struggle with particular types of differences.

### 4.2.2 ANNOTATING ACTION DIFFERENCES

For each action $a_j$ and its corresponding differences, annotators reviewed video pairs $(v_A, v_B)$ side-by-side, with the ability to step through frames. Each difference was labeled as 'A' if it applied more to video $v_A$, 'B' if it applied more to $v_B$, or 'C' if the difference was insignificant. Consistent annotation was achieved by assigning a single annotator to each action, ensuring that models are evaluated uniformly across all samples. This avoids the *calibration* challenge, that different annotators may have different thresholds for significance.

To verify annotation quality, a second annotator reviewed 25% of the samples. We assessed disagreements where one annotator marked 'A' and the other marked 'B', which occurred in only 2% of cases, indicating low error rates. Annotators were provided with clear visual guidelines to ensure accurate and impartial labeling. On average, annotators spent three minutes per video pair to evaluate about eight differences.

### 4.2.3 ANNOTATING DIFFERENCE LOCALIZATIONS

In addition to action differences, VidDiffBench provides localization annotations, pinpointing the exact frames in each video where key differences occur. Since identifying localizing frames and aligning them across videos is a key step in performing video action differencing, these annotations enable analysis of model weaknesses, for example through ablation tests in our results section.

We define specific *key points* for each action, representing critical frames where important movements occur. For example, in a squat, key points might include "knees start to bend" and "reaches lowest position." Differences are then linked to these key points: for example the difference "faster squat descent" is defined as the frame spanning "knees to bend" and "reaches lowest position". Further details are provided in Appendix C.2.

### 4.3 DATASET SPLITS AND STATISTICS

**Dataset Splits**  To account for varying levels of difficulty in VidDiffBench, we categorize actions into *easy*, *medium*, and *hard* splits. GPT-4o was used to assign actions to these splits based on descriptions, difference lists, and video lengths. The easy split includes simple movements like Fitness exercises, while medium and hard splits contain more complex actions like Ballsports, Diving, Music, and Surgery. This ensures that models are challenged across a range of difficulties, from basic movements to subtle, fine-grained comparisons.

**Dataset Statistics**  VidDiffBench includes 549 video pairs, 4,469 annotated differences, and 2,075 key point annotations across Fitness, Weightlifting, Ballsports, Surgery, Music, and Diving domains. Video lengths range from a few seconds to several minutes, providing comprehensive coverage of different action complexities. This diversity ensures that VidDiffBench is a robust benchmark for testing and advancing models in fine-grained action comparison. Under the closed setting, the A/B ratio is 0.493/0.507, and in the open setting, the A/B/C ratio is 0.259/0.264/0.476.

## 5 VIDDIFF METHOD

We propose a three-stage framework, the VidDiff Method, that addresses the Video Action Differencing task. The method follows an agentic workflow (Anthropic, 2025) consisting of three components: Difference Proposer, Frame Localizer, and Action Differencer Figure 2. The stages decompose the differencing task into logical steps, and leverage strong zero-shot models for each step. The method described is for the open setting. The method for the closed setting is the same, except the LLM query for candidate differences in stage 1 is replaced with the ground truth differences.

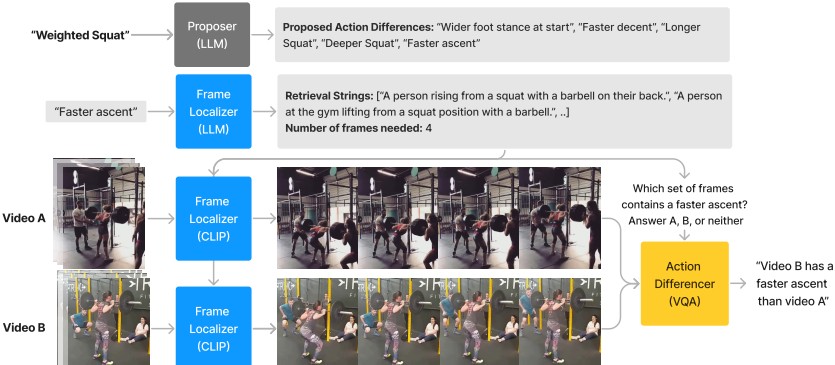

Figure 2: VidDiff Method. One input is an action description (e.g. "weighted squat"). The Difference Proposer generates potential differences using a large language model (LLM). The Frame Localizer assigns frames where these differences are observable. Finally, the Action Differencer checks each difference using a vision-language model, determining whether it applies more to video A or video B, or neither.

**1. Difference Proposer:** The Difference Proposer module generates candidate differences for a given action description $s$. It leverages the extensive knowledge embedded in large language models (LLMs) to predict likely differences between the two videos. For example, given the description "A practice basketball jump shot", the module might generate difference candidates such as "the athlete jumps higher". These difference statements, which are visually assessable, form the basis for further analysis. The goal of this stage is to create a diverse set of meaningful and relevant comparisons.

**2. Frame Localizer:** The Frame Localizer module focuses on identifying the most relevant frames in the video where the proposed differences can be observed. By retrieving the most salient segments from both frames, we solve the key challenge of *temporal localization of sub-actions*, which makes the next stage more effective. Our approach is to do temporal sub-action segmentation. The LLM takes uses the action description string to produce a list of sub-actions, along with retrieval strings to guide localization. A pretrained CLIP model (Radford et al., 2021) is used to compute frame similarity based on these retrieval strings, and then we assign each frame to one of the sub-actions. Here, we use a Viterbi-based algorithm (Kukleva et al., 2019), which assigns each frame to a sub-action based on its similarity score, while enforcing that the frames follow the fixed sequence of sub-actions. Finally, the LLM predicts a mapping between the sub-actions and their corresponding differences, yielding a set of precisely localized frames for each difference.

**3. Action Differencer:** In the final stage, the Action Differencer module validates the proposed differences using vision-language models (VLMs). Given the localized frames from both videos, this module poses multiple-choice questions (derived from the generated difference candidates) to a VLM, which determines whether each difference is more pronounced in $v_A$, $v_B$, or if it is indistinguishable. This stage transforms the problem into a structured multiple-choice task. Moreover, by providing the localized-frames relevant to each difference.

Overall the VidDiff method is structured to localize the key parts of the video where differences are possible, which should make visual comparison with the VLM easier.

## 6 RESULTS

In this section, we present the results of evaluating large multimodal models (LMMs) and our VidDiff Method and on the challenging task of video action differencing on our VidDiffBench benchmark. Our experiments show the complexity of this task, particularly in capturing subtle, fine-grained action differences across diverse video categories. We demonstrate that existing state-of-the-art LMMs, such as GPT-4o and Gemini, struggle with these challenges, while our proposed VidDiff Method outperforms the baselines, especially in the close-set evaluation. Through detailed error analysis and ablation studies, we uncover key factors that influence model performance, shedding light on future directions for improving video-based model capabilities.

## 6.1 MAIN RESULTS

As described in Section 3.2, we evaluate our approach on both the *closed-set* and *open-set* tasks. In the closed-set task, models are provided with predefined difference descriptions and must predict whether the difference applies to video $A$ or $B$. In the open-set task, models are tasked with both generating the difference description and making a prediction. These tasks are fundamental to assessing models' capabilities in fine-grained action comparison.

For our experiments, we benchmark large multimodal models (LMMs) that have demonstrated strong performance in video tasks. Specifically, we use top models from the Video-MME benchmark (Fu et al., 2024): GPT-4o (Achiam et al., 2023), Gemini-1.5-Pro (Reid et al., 2024), Claude 3.5 Sonnet Anthropic (2024), and the leading open-source models, Qwen2-VL-7B (Wang et al., 2024; Bai et al., 2023) and LLaVA-Video (Zhang et al., 2024). Following model guidelines, we provide Gemini, Qwen, and VideoLLaVA with video inputs, while for GPT-4o and Claude we give frames, with text prompts explaining which frames belong to which video. For categories with shorter, fine-grained actions (e.g., Fitness, Ballsports, and Diving), we sample frames at 4-6 fps, while for longer actions (e.g., Music and Surgery), we sample at 2 fps. Our method, VidDiff, is evaluated alongside these baselines, were the proposer LLM is `gpt-4o-2024-08-06`, the localizer embedding model is `CLIP ViT-bigG-14`, and frame differencer VLM is `gpt-4o-2024-08-06`. The results are results shown in Table 2 and Table 3.

**Closed-Set Benchmark Performance**   The closed-set results are in Table 2, revealing that video action differencing is a highly challenging task. While some models surpass the random-guessing baseline of 50% – where gray shading indicates better-than-random with statistical significance – their improvements are modest, especially in the harder splits where no model performs significantly better than chance. Gemini, which has emphasized its results in video understanding, has the strongest overall performance. Our VidDiff Method, which uses GPT-4o as a visual perception backbone, outperforms GPT-4o on the raw video frames and is second overall, demonstrating the value of our scaffolding for this task. LLava-Video is competitive with GPT and Claude, while Qwen2-VL performs poorly, possibly related to instruction-following challenges appendix G.4

Table 2: Results for closed setting (accuracy). Best scores in **bold**, second best underlined. Scores are better than random, with statistical significance highlighted in gray. Significance is p-value< 0.05 on a binomial test.

|  | Easy | Med | Hard | Avg |
|---|---|---|---|---|
| **GPT-4o** | 58.3 | 53.2 | 48.9 | 53.5 |
| **Gemini-1.5-Pro** | **67.8** | 53.6 | 51.7 | **57.7** |
| **Claude-3.5-Sonnet** | 57.1 | 50.5 | **52.5** | 53.4 |
| **LLaVA-Video** | 56.6 | 52.0 | 48.3 | 52.3 |
| **Qwen2-VL-7B** | 49.0 | 52.6 | 49.6 | 50.4 |
| **VidDiff (ours)** | 62.7 | **56.2** | 50.0 | 56.3 |

**Open-Set Benchmark Performance**   In the open-set task (Table 3), our method outperforms all other models across most splits, except on the medium difficulty. Among the LMMs, GPT-4o performs much better than Gemini. We analyze this gap by breaking down errors into two categories: *difference recall error*, where the model fails to generate the ground-truth difference, and *flipped prediction error*, where the generated difference is correct but the prediction ('A' or 'B') is incorrect. The closed-set results show that Gemini has lower flipped prediction error, suggesting that Gemini's main weakness is in difference recall. Specifically, on the easy split, Gemini's recall error is 66% compared to GPT-4o's 30%. Despite generating a similar number of differences as GPT-4o, Gemini struggles to identify the most important ones in our taxonomy, which hampers its performance. Success in the open setting requires strong language capabilities, and this limitation is the bottleneck for handling subtle differences. This explains why, when using the same language proposer, our model performs similarly to GPT-4o.

Table 3: Results for open setting (recall@$N_{\text{diff}}$). Best scores in **bold**, second best underlined.

|  | **Easy** | **Med** | **Hard** | **Average** |
|---|---|---|---|---|
| **GPT-4o** | 45.7 | **41.5** | 38.0 | 41.7 |
| **Gemini-1.5-Pro** | 30.3 | 30.5 | 24.1 | 28.3 |
| **Claude-3.5-Sonnet** | 37.8 | 34.6 | 34.3 | 35.6 |
| **LLaVA-Video** | 7.8 | 9.0 | 8.5 | 8.4 |
| **Qwen2-VL-7B** | 11.2 | 8.8 | 1.6 | 7.2 |
| **VidDiff (ours)** | **49.9** | 37.9 | **38.5** | **42.1** |

## 6.2 ABLATION STUDIES

We conducted ablation studies to better understand the individual contributions of different components within VidDiff. These studies focus on the Closed setting, isolating the effects of the frame differencing and frame localization stages.

**Frame Differencer Image Comparison** In the final stage of VidDiff, the model performs visual question answering (VQA) on frames retrieved from the two videos. To evaluate the effectiveness of this process, we conducted a test using the ground-truth timestamp annotations from VidDiffBench. The results (Table 4) show that even with perfect frame alignment, zero-shot VLMs struggle to consistently detect subtle differences in images. Performance decreases significantly on the medium and hard splits, which suggests room for improvement in zero-shot VLMs' image understanding capabilities.

**Frame Localization Design** We also analyzed the performance of the Frame Localizer in the closed-set case for the easy split, using ground-truth difference proposals to measure VQA accuracy. Table 5 shows that random frame retrieval leads to significant performance drops, while the addition of Viterbi-based decoding (which enforces a fixed action transcript) substantially improves accuracy. The improvement suggests that temporal alignment plays a critical role in achieving robust video differencing.

| Split | Easy | Medium | Hard |
|---|---|---|---|
| Acc | 78.6 | 61.2 | 51.0 |

Table 4: Ablation study results for frame differencing VQA with ground truth frames. Questions are 3-way multiple-choice.

In summary, these ablation studies confirm that both accurate frame localization and careful VQA processing are essential to achieving strong performance in video action differencing.

## 6.3 DIFFERENCE-LEVEL ERROR ANALYSIS

VidDiffBench's predefined taxonomy allows us to analyze model performance on 148 specific types of action differences, highlighting where models succeed and fail. The results for each difference are detailed in Appendix Table 14, and we perform a statistical significance test to compare models against the random-guessing baseline.

| Method | Accuracy |
|---|---|
| Oracle (GT timestamps) | 78.6 |
| Random | 50.1 |
| Ours w/o Viterbi Decoding | 57.4 |
| Ours | 62.7 |

Table 5: Ablation on frame localization using different retrieval techniques on easy.

We find that model performance is highly dependent on the visual complexity of the action and the difficulty of localization. Successful examples (Figure 3, left column) show high accuracy for simple, easily localized actions, such as "wider foot stance" in hip rotations (83% accuracy) or "guiding the ball" in a basketball layup (90% accuracy). These cases feature coarse differences that are apparent in most frames, or require only approximate localization.

Conversely, failure cases (Figure 3, right column) often involve precise localization or fine-grained differences. For instance, identifying the angle of a diver's entry into the water in a 10m dive' requires frame-perfect alignment, and recognizing subtle changes in speed in "piano scales" is difficult

when reasoning over multi-frames. These challenges highlight the limitations of current models in handling fine-grained video analysis.

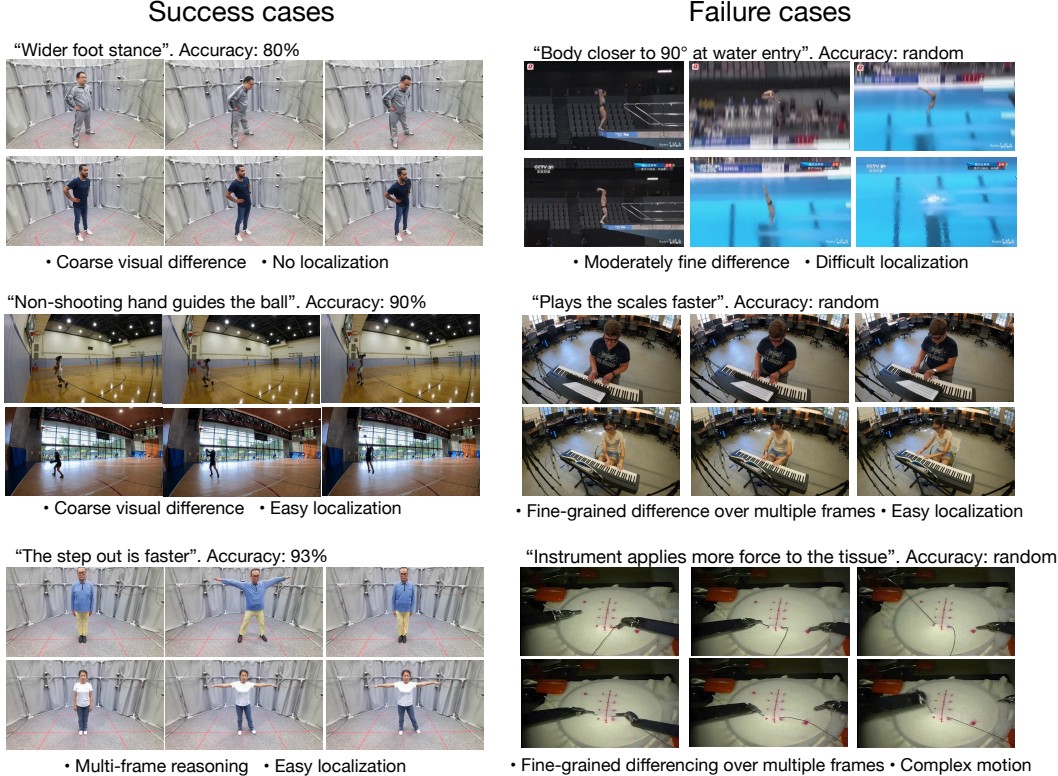

Figure 3: Examples of 'success cases' (left) – differences where GPT-4o has high accuracy – and failure cases (right). Success cases typically involve coarse differences, easy localization, or simple actions, while failure cases often involve fine differences, precise localization or complex actions.

## 7 CONCLUSION

In this paper, we introduce the novel task of Video Action Differencing (VidDiff), aimed at comparing actions in videos. We define this task, compile a meticulously annotated benchmark, and propose a zero-shot agent-based framework. Our findings demonstrate that this task is feasible with current foundation models, although more challenging splits in the benchmark reveal significant opportunities for further methodological improvements. We believe that Video Action Differencing represents a promising research direction with broad applications in fields such as skill acquisition, sports analytics, and scientific research.

## 8 FUTURE WORK AND LIMITATIONS

While our work demonstrates the potential of Video Action Differencing, there are areas for future improvement. Enhancing frame retrieval techniques could improve performance. While many large models have emergent capabilities not trained for (Tang et al., 2023a; Burgess et al., 2024), explicit training for comparing fine-grained features of Vision-Language Models (VLMs) is likely underutilized. Further, developing methods tailored to specialized domains such as healthcare or education could unlock more targeted applications. Limitations in our current approach include reliance on general foundation models, which may struggle with domain-specific tasks or fine-grained comparisons. We hope this work encourages further exploration into broader video comparison methods and inspires advancements in these areas.

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

## A  BENCHMARK: DOWNLOAD INSTRUCTIONS

Our benchmark is released at https://huggingface.co/datasets/jmhb/VidDiffBench. It has complete instructions on how to access annotations, how to download external datasets, and all licenses for our annotations and the source video datasets.

## B  BENCHMARK: DIFFERENCE ANNOTATION TAXONOMY GENERATION

Each dataset underwent a thorough taxonomy generation process. Details for each dataset are presented in this section. Most datasets were first processed with the Difference Proposer.

### B.1  GENERATING JIGSAWS DIFFERENCE ANNOTATION TAXONOMY FOR SURGERY VIDEOS

To produce the JIGSAWS taxonomy for surgery videos, we first used the Difference Proposer to generate difference candidates. As we found many proposed variations to be irrelevant, we consulted a surgeon from ¡anonymous_hospital¿. We first showed them a variety of videos from the JIGSAWs dataset and brainstormed visually discernible differences. Next, we shared the GPT-generated variations with the surgeon and discussed which ones should be added to the brainstormed listed. Lastly, we dropped all differences that could not be annotated consistently. For instance, we removed "Surgeon exploits robotic instrument's range of movement more efficiently in Video A than in Video B", as the difference is subject to interpretation.

### B.2  GENERATING EGO-EXO4D DIFFERENCE ANNOTATION TAXONOMY

For datasets with expert annotation, such as Basketball, Soccer, and Music, we processed the expert commentary with GPT instead of asking for difference proposals directly. We asked GPT to summarize the expert commentary using the following prompt:

```
Below are a sequence of text strings.
The strings are written by experts who are watching videos of a task with this
    description: "{action_descriptions}".
As the experts watch each video, they pause the video and record verbal commentary about
    how that person is performing the task.

Return a list of text strings that summarizes what are the key visual cues that the
    expert is looking when they provide feedback.
Each item in the list should be specific and testable, so that anybody watching a single
    video can assess whether that visual cue applies to a particular video.

Your response should be a json with this structure:
{ "summary_texts" : ["text0", "text1", ...] }
The list should have at least 15 items.

Here are the texts to summarize:
{texts}
```

As GPT has no knowledge of the specifics of the dataset, we then manually parsed the proposed list and kept items that could be distinguished by non-experts based on visual information only. Some differences were not visible in our data. For instance, the "wrist snap" in basketball was excluded, as it could not be discerned from the videos.

As we found that GPT lost a lot of information from the expert commentary, we also manually parsed the expert comments and added key visual cues that were mentioned by the experts.

### B.3  GENERATING FINEDIVING DIFFERENCE ANNOTATION TAXONOMY

For the diving dataset, we used the Difference Proposer. As all videos are of experts, there is little variation between the videos. We thus used the diving score annotations given in the dataset, to find pairs of images with more variability and used those proposed differences that were discernible in these pairs.

## C  BENCHMARK: ANNOTATION DETAILS

Two annotators were provided with a list of variations and a folder containing all videos as well as concatenated videos of two actions next to each other. They were instructed to annotate differences only as such, if they were obvious. The instructions for the annotators were as follows:

```
You'll get pairs of videos and be asked questions about how they're different. They'll
    be specific querstions:
- E.g. "Which one has a wider foot stance: (a) video A, (b) video B, or (c) they're the
    same.
- if it's hard to tell whether there really is a difference, then say "c". Rule of
    thumb: once you've found the important point in the video, if it takes you more
    than 10 seconds to make a decision, then say "c".
```

### C.1  BENCHMARK: THE DIFFERENCE TAXONOMY

The difference taxonomy is available as part of the benchmark release at this link. The full list of differences can also be previewed in the analysis table 14.

### C.2  BENCHMARK: RETRIEVAL ANNOTATION GENERATION

For the closed-evaluation scenario, we need to temporally align the videos we wish to compare. For this alignment, we annotate retrievals, which are important and identifiable moments within an action. To generate retrievals, we used the Frame localizer, where we prompt GPT4-V to propose stages for a given action (see Section C.3 for details). We found that the stages were sometimes to coarse. We thus either manually identified key moments that helped retrieve sections of the videos important for comparing differences, or, for JIGSAWS consulted our expert. The retrieval annotations are available as part of the VidDiffBench benchmark release, and are in the folder called 'retrieval'.

### C.3  PROMPTS FOR ACTION ASSIGNMENT TO THE EASY/MEDIUM/HARD SPLITS

```
I'm designing a benchmark for comparing pairs of videos of the same action.
We have many actions and each action has a list of differences we look for.
The benchmark's task is to examine differences and say whether the statement applies
    more to "video A" or "video B".

Below we show a dictionary where each element is a single action.
Each action has an "action_description" describing the action.
It also has "average_seconds_per_video", for the median length of videos in seconds.
Each action has a dictionary of "differences", where each difference has these keys:
- 'name' for the difference
- 'description' describing the difference

Finally, for each action, there are two unfinished options:
- 'split' which currently says 'easy|medium|hard'
- 'split_reason' which currently says '...'
Your task is to fill in these values:
- Decide whether the 'split' value is 'easy', 'medium' or 'hard'. This evaluation judges
     the difficulty of performing actionn difference comparison for all differences
    within an action. Having a high number of actions should not be considered as
    criteria for difficulty.
- Justify your choice in 'split_reason'.

Return the same dictionary as json, with the values of 'split' and 'split_reason'
    populated.
Here are the actions.
{actions}
```

The `actions` field is replaced with a json with the structure that is described in the prompt.

## C.4 DIFFICULTY SPLITS

The result of the difficulty splits is in table 6.

Table 6: The difficulty splits with action code and names

| Split | Action | Action Name | Action description |
|---|---|---|---|
| easy | fitness_0 | Hip circle anticlockwise | fitness exercise called standing hip circle with hands on hips, one rotation anticlockwise |
| easy | fitness_3 | lucky cat | fitness exercise called two arm standing lucky cat starting with arms up, one repetition |
| easy | fitness_4 | Squat Knee Raise side view | squat without weights, then knee raise on left side |
| easy | fitness_6 | Hip circle clockwise | fitness exercise called standing hip circle with hands on hips, one rotation clockwise |
| medium | ballsports_0 | basketball jump shot | a person is doing a basketball mid-range jump shot, starting with the ball in their hand, no defense, practice only |
| medium | ballsports_1 | basketball mikan layup | a person does the Basketball drill called the Mikan layup where they start under the basket, do a layup with the right hand and catch it, do a left hand layup and catch it, no defense, practice only |
| medium | ballsports_2 | basketball reverse layup | a person playing basketball does a reverse layup starting from the left side of the basket and lays it up with their right hand on the right hand side, no defense, practice only |
| medium | ballsports_3 | soccer penalty kick | a person does a soccer drill where they do a single penalty kick, practice only, no defense, no goalie |
| medium | diving_0 | diving | competitive diving from 10m |
| medium | fitness_1 | Opening and closing step left side first | fitness exercise called opening and closing step on left side and then opening and closing step on right side |
| medium | fitness_2 | car deadlift | a single free weight deadlift without any weight |
| medium | fitness_5 | Squat Knee Raise diagonal view | a squat then a knee raise on left side |
| medium | fitness_7 | Opening and closing step right side first | fitness exercise called opening and closing step on right side and then opening and closing step on left side |
| hard | music_0 | piano | a person is playing scales on the piano |
| hard | music_1 | guitar | a person is playing scales on the guitar |
| hard | surgery_0 | Knot Tying | The subject picks up one end of a suture tied to a flexible tube attached at its ends to the surface of the bench-top model, and ties a single loop knot. |
| hard | surgery_1 | Suturing | The subject picks up needle, proceeds to the incision (designated as a vertical line on the bench-top model), and passes the needle through the fake tissue, entering at the dot marked on one side of the incision and exiting at the corresponding dot marked on the other side of the incision. After the first needle pass, the subject extracts the needle out of the tissue, passes it to the right hand and repeats the needle pass three more times. |
| hard | surgery_2 | Needle Passing | The subject picks up the needle (in some cases not captured in the video) and passes it through four small metal hoops from right to left. The hoops are attached at a small height above the surface of the bench-top model. |

## C.5 VALIDATING SPLIT GENERATION - HUMAN STUDY

Choosing the difficulty splits requires a holistic view of all the actions, so we decided it didn't make sense for experts to suggest them, since they are only familiar with a few actions each. On the other hand, we didn't want to rank the splits based on performance of current models since this felt like biasing towards current models; and besides, the performance for many actions in 'medium' and 'hard' is already random, so it would be hard to differentiate these actions. LLMs are a good candidate because they have a good understanding of the actions and are relatively free of the biases of this paper's authors Furthermore, human annotators could not do the ranking, because no human annotated all the actions.

To further support the choice of an LLM, we asked 3 humans to rank the action comparisons from easiest to hardest, and compared against the LLM ranking. We then computed the Spearman's rank correlation between all ranking sets, and the results are in table 7. The mean of the pairwise correlations between the humans was 0.602, while the mean of pairwise correlations between the LLM and humans was higher at 0.673. This shows (i) that there is non-negligible variability in human rankings, and (ii) that the LLM ranking is reasonable, and actually better correlated with most humans compared to several of the human annotations.

Table 7: Results on human evaluation study on choosing the splits. This is the Spearman's rank corrlation between the ranks of action dificulty, comparing our LLM approach and 3 humans.

|  | LLM | Human 1 | Human 2 | Human 3 |
|---|---|---|---|---|
| **LLM** |  | 53.1 | 68.0 | 80.6 |
| **Human 1** | 53.1 |  | 45.9 | 64.5 |
| **Human 2** | 68.0 | 45.9 |  | 70.3 |
| **Human 3** | 80.6 | 64.5 | 70.3 |  |
| **Average** | 67.3 | 54.5 | 61.4 | 71.8 |

## C.6 FURTHER DATASET CONSTRUCTION CONSIDERATIONS

**Camera angles**   The change of camera angle perspective does make the task harder. For samples in the 'Fitness' category, the camera angle is the same because the source dataset has a fixed camera rig, and we chose to use the same camera angle. For samples in 'diving' and 'surgery' categories, the camera angle is approximately the same. On the other hand, samples from 'ballsports' and 'music' categories can change. A related attribute (not mentioned here) is differences in background – similarly the 'ballsports' and 'music' categories often had different backgrounds as well. Importantly, these attributes were considered when assigning the difficulty splits. This may partly explain why the fitness exercises are all in the easy and medium split.

**FPS**   Each video pair has the same fps. In case others want to leverage our code with new videos, our code does handles the case where FPS is different. Specifically, the input configuration has a value for the target FPS for running inference, and we subsample the video to have this FPS. (If the videos cannot be subsampled to have the exact target fps, then a warning is printed).

**Impact of different actor heights**   For annotator instructions, we clarified that all differences on things like distance should be relative to the actor's height. We gave the example of 'wider foot stance', saying that if a 5ft actor and a 6ft actor both had their legs 3ft apart, then the shorter actor has a 'wider foot stance' relative to their height. This reflects what is commonly understood by descriptions like these in skills coaching.

## D   BENCHMARK STATISTICS

Beyond the main statistics in the main, table 8 shows further statistics broken down by difficulty splits.

Table 8: Detailed data statistics by split

| Split | easy | medium | hard | Overall |
|---|---|---|---|---|
| **# video pairs** | 95 | 265 | 197 | 557 |
| **Avg video length (secs)** | 2.1 | 3.9 | 18.7 | 8.8 |
| **Total video length (mins)** | 6.5 | 34.7 | 122.5 | 163.7 |
| **# differences tagged** | 1224 | 4788 | 3542 | 9554 |
| **StdDev within retrieval type** | 8.4% | 5.2% | 4.1% | 5.9% |
| **StdDev across retrieval types** | 17.3% | 25.7% | 20.2% | 21.0% |
| **Difference annotations count** | 578 | 1771 | 2370 | 4719 |
| **Difference annotations A/B/C distribution** | 167/190/221 | 622/605/1143 | 435/452/884 | 1224/1247/2248 |

Average video length is longer as the difficulty gets higher: 2.1/3.9/18.7 seconds, for easy/medium/hard. Compared to video QA datasets, the lengths are relatively shorter because we focus on fine-grained action understanding in how actions are performed. The total length of videos is 163 minutes.

**Retrieval tags, temporal bias**   For the 'retrieval tags', we first show the number of retrieval tags – 9554 total. To give insight into their distribution within each video, each instance is normalized to the video length, and compute its 'video location'. E.g. in a squat, the starting position might be position 0.1, the bottom of the descent 0.45, and the squat finish at 0.87. Within each retrieval type, we compute 'StdDev within retrieval type', which intuitively measures how well-aligned are

the key points in the video. For example, if the average squat video records 'bottom of descent' at location 0.45, and 'within StdDev' is 0.06, then the mean distance from the average is 0.06 (so at 0.39 or 0.51). The 'within StdDev' is on average 0.059, indicating there is some variation in retrieval position, but there is temporal bias. This is expected since each video is trimmed and contains an atomic action. Future benchmarks could use untrimmed videos to make retrieval annotations less aligned, but the present benchmark is already difficult for SOTA models, so this is unnecessary now.

**Retrieval tags, coverage**   We also measure 'StdDev across retrieval types', meaning the standard deviation of different retrieval classes within one video. Intuitively this measures how much of the video is 'covered' by retrieval keypoints. This is 0.21 on average. So if the mean of retrieval keypoints were 0.5, then the average retrieval annotations is around 0.29 or 0.71 in the video.

**A/B/C distribution**   Additionally, we have shown the count of difference annotations and the A/B/C distribution; the 'no difference' annotation of 'C' is the most prevalent.

## E    BENCHMARK: RELATED VIDEO PAIR DATASETS

A small number of prior works have datasets of paired videos with some label of the difference. However none have labels for fine-grained comparison while also having a large scale.

- Nagarajan & Torresani (2024) has a large-scale dataset of video differences in instructional video – large enough to be used for instruction tuning. However their differences are very coarse-grained, for example 'cooking video A forgot to add salt'. Here, large-scale is possible because differences can be derived automatically from annotated instructional video datasets.
- (Balakrishnan et al., 2015) considers fine-grained action differences. However their dataset is very small (less than 50), and they have no labels.
- (Doughty et al., 2018) has a dataset of paired actions called EPIC-Skills2018. Here, the scale is large, but the difference label is more coarse: a binary for which video shows more skill.

## F    EVALUATION

The evaluation code is available at this GitHub repo https://github.com/jmhb0/viddiff.

### F.1    CLOSED EVALUATION DESIGN CHOICE

Our closed evaluation setting has options for 'A' and 'B', but not 'c'. Our initial approach to formulating this did include an option 'C' for insignificant differences. However, the challenge of calibration made fair evaluation difficult. For example, when comparing two videos of a basketball shot to evaluate stance width, the question arises: how different is "different enough" to be both relevant for skill learning and perceptible? Different annotators may apply varying thresholds for what constitutes a significant difference, leading to inconsistencies. Introducing option 'C' further complicates evaluation because it requires calibrating not only the human annotators but also the VLMs, which may have different internal thresholds for perceiving significance. To address these challenges, we adopted the following approach:

- Annotators were instructed to choose either 'A' or 'B' only when the difference was clearly perceptible.
- We limited the evaluation of VLMs to cases where there was a very clear ground truth answer of either 'A' or 'B'.

This method ensures fairness by focusing on scenarios with unambiguous ground truth, avoiding complications introduced by subjective calibration thresholds. While we briefly discuss this in the section on annotation creation, we recognize that this is a nuanced point.

## F.2 Evaluation: matching with LLMs

As described in section 3.2, we use an LLM

We use 'gpt-4o-2024-08-06' with the following prompt for matching ground truth to predicted descriptions. We do one prompt per video pair sample.

```
You are analyzing videos of people performing a specific action described as "{
    action_description}."

In this task, a "difference" refers to how two people might perform the same action in
    distinct ways.
Each difference consists of:
- A name (key) and
- A description that explains how the two performances differ visually.

You are provided with two dictionaries:
- Dictionary 0: {differences0}
- Dictionary 1: {differences1}

Your task is to match the differences from Dictionary 0 to Dictionary 1. Here's how:
1. For each entry in Dictionary 0, find the best match in Dictionary 1.
2. Each item in Dictionary 1 can only be used once.
3. Only match entries if their description strings are visually similar, even if the
    word choices differ. If no suitable match exists, return "None."

Your output should be a JSON object where:
- The keys are from Dictionary 0: {dict0_keys}.
- The values are a key from Dictionary 1 or "None" if no match is found.
- The available keys from Dictionary 1 are: {dict1_keys}.

Example output format:
{
    "0": "3",
    "1": "None",
    "2": "1",
    "3": "5",
    ...
    "{final}" : "0"
}
Important: the keys in this dictionary should be" {dict0_keys}
```

We replace:

- {action_description} with a string describing the action
- {differences0} with a dictionary where keys are gt difference keys ("0","1",...
  and values which are strings describing differences.
- {differences1} is the same as {differences0}, except for the predicting differences.
- {dict_keys} are the keys in {differences0}
- {dict1_keys} are the keys in {differences1}
- {final} is the highest key in {differences0}

An issue with matching is that the prediction actually has the opposite value to the ground truth – e.g. "the arms are more straight" vs "the arms are more bent". If this is the case, then the prediction 'a' should be reversed to 'b' and vise versa. To identify those cases, we use this prompt, and we evaluate them in batches of 6 difference sequences at once.

```
Task:
You will be given pairs of statements. Your task is to determine the logical
    relationship between each pair.
```

```
Instructions:
1. Analyze Each Pair: For each pair of statements, carefully analyze their meaning and
      relationship.
2. Categorization:
    - Return "0" if the statements are equivalent, very similar, or differ only in minor
        details.
      Example: "X is bigger than Y" and "X is larger than Y" should both return "0".
    - Return "1" if the statements are direct opposites in meaning.
      Example: "X is bigger than Y" and "X is smaller than Y" should return "1".
3. Edge Cases:
    - Avoid returning "1" for statements that are not true opposites, even if they have
        some differences in detail or degree.
      Example: "X is much bigger than Y" and "X is slightly bigger than Y" should still
          return "0".

Output Format:
- Your response should be a JSON object with a single key "results" and an array of
      string values "0" or "1" as its value.
- The array should exactly match the number of statement pairs given in the input.

Input Format:
- The list of statement pairs will be provided in the following format:

{statements}

Important Requirements:
- Ensure that each value in the output array is either "0" or "1".
- The length of the "results" array must exactly match the number of input pairs.
```

We replace the {statements} with a list, where each element is a two element list of strings, which are matched difference descriptions.

## F.3   PROMPTS FOR LMM BASELINES

The LMM baselines – GPT, Gemini, and Qwen – all receive the same prompts. Prompts for action assignment to the easy/medium/hard splits:

```
Here are two videos of an action with the following description: "{action_description}".
{video_representation_description}

Below is a set of identified differences that describe how the action be performed
    differently.
Each difference is associated with a unique key:
{differences_annotated}

Your task is to predict, for each difference, whether it is more true for video 'a' or
    video 'b'.
{target_out}
```

The {action_description} is replaced with a string describing the action which is the same as in table 13. The {differences_annotated} is a dictionary mapping the ground truth difference key to a difference description, and are the same stringes used in table 14. The {video_representation_description} tells the model how the video data is passed in. If 2 videos – for Gemini and Qwen – then it's:

```
We have passed 'video a' and 'video b' as videos to the prompt in that order.
```

If passing in the videos as frames – for GPT – then it's:

```
We have passed a sequence of images into the prompt.
The first {vid0_nframes} are video a. The last {vid1_nframes} are video b.
The frame rate is the same and is {fps}.
```

The open prompt is:

```
Here are two videos of an action with the following description: "{action_description}".
{video_representation_description}

Return a list of 'differences' in how the action is being performed.
Each difference should have a 'description' that is a specific statements that is more
    true in one video compared to the other video.
Then there is a 'prediction' which is 'a' if the statement applies more to video a and '
    b' if it applies more to video b.

The difference descriptions should be visual and about how the action is performed.
For example 'the jump is higher', or 'the arm is more straight'.

The difference descriptions should not refer to a specific video.
For example you would not say 'the jump in video B is higher'.
Instead, the 'description' would be 'the jump is higher', and the 'prediction' is 'b'.

Suggest no more than {n_differences} differences.

Return a json like this, replacing '...' with actual content:
{
    "0" : {
            "description" : "....",
            "prediction" : "a|b"
        },
    "1" : {
            "description" : "....",
            "prediction" : "a|b"
        },
    ...
}
```

## F.4 VALIDATION OF MATCHING PROCESS

We leverage LLMs in open evaluation to identify matching between the ground truth difference description strings and the predicted differences. Here we validate that this is a reasonable approach.

**Robustness to multiple LLM runs**  The LLM evaluation is robust to random seed. We repeated the evaluation five times with different random seeds and observed a standard deviation of only 0.7 in the final evaluation score. This indicates that the results are consistent across runs. Although the prompt was specifically engineered for the GPT-4o-2024-08-06 model, we ensured consistency by fixing the model for all evaluations, treating all comparisons under identical conditions.

**Comparison with Human Evaluation**  To measure alignment with humans, we recruited 3 human annotators to perform open evaluation matching, each with 44 video pairs and 347 individual differences. For each video pair, they were provided with a list of ground truth differences, and asked to match each one to a predicted difference from a list, or to suggest no match. We calculated inter-rater agreement across annotators and the automated LLM system. The results are in table 9. We can see semantic matching proved to be challenging for humans – the mean of pairwise rater agreement from each human to the other humans was 75.7%. Meanwhile, the mean agreement between our automated system and human annotators was 73.9%. Therefore, our LLM-based approach is on par with human annotators, while being completely automatic.

**Details of Prompt for LLM Evaluation**  The LLM prompt was carefully developed using a prompt engineering workflow. We selected a set of four evaluation samples, covering two actions and two models, and iteratively refined the prompt based on performance in individual runs. For example, we added the instruction: "Only match entries if their description strings are visually similar, even if the word choices differ." This adjustment was necessary because the LLM struggled to match equivalent descriptions phrased differently (e.g., "the feet stance is wider" vs. "the legs

Table 9: Agreement rate of LLM and human predictions for the evaluation matching.

|  | LLM | Human 1 | Human 2 | Human 3 |
|---|---|---|---|---|
| **LLM** |  | 72.4 | 74.0 | 70.1 |
| **Human 1** | 72.4 |  | 75.0 | 78.2 |
| **Human 2** | 74.0 | 75.0 |  | 73.9 |
| **Human 3** | 70.1 | 78.2 | 73.9 |  |
| **Average** | 72.2 | 75.2 | 74.3 | 74.0 |

are spread wider apart"). While this approach achieved satisfactory results, we acknowledge that the prompt could be further optimized using more systematic methods, such as DSPy Khattab et al. (2024). Exploring such techniques is a promising direction for future work.

# G  RESULTS: MORE ANALYSES

## G.1  RESULTS: ACTION-LEVEL MODEL COMPARISON

We have performed a more thorough comparison of the different state-of-the-art LMMs on VidDIff-Bench, added a small subsection to the results, and a discussion in appendix. Specifically we look at each action, and compare the different LMMs.

First, we show the correlations in the per-action scores between models in table 10

Table 10: Correlations between models where the data is the action-level accuracy.

|  | GPT | Gemini | Claude | LLava-Video | Qwen2-VL |
|---|---|---|---|---|---|
| **GPT-4o** |  | 0.152 | 0.375 | 0.243 | 0.273 |
| **Gemini-1.5-Pro** | 0.152 |  | 0.215 | 0.111 | 0.223 |
| **Claude-3.5-Sonnet** | 0.375 | 0.215 |  | 0.261 | 0.220 |
| **LLaVA-Video** | 0.243 | 0.111 | 0.261 |  | 0.376 |
| **Qwen2-VL-7b** | 0.273 | 0.223 | 0.220 | 0.376 |  |

The correlations are generally low, but there are 3 clusters of models. LLaVA-Video and Qwen-2-VL are in a cluster; they are both open-source, and have the same LLM backbone. Then GPT-4o and Claude-Sonnet cluster together, and Gemini is not similar to any other model. We can speculate that for video data, Claude and GPT have similar training strategies, while Gemini's is different.

Next we compare model performance within one action, and this is two large tables. table 11 is the action-level performance of each model. Then table 12 is the 'relative performance': the difference between the model score on that action compared to the mean score across all models for the action. The most significant results in the benchmark are on the easy split. Here, the improvement in score is uniform for all models. The models generally close perform similarly each other. The relative performance is usually less than 10 points – when it is higher, the sample size is very small.

By comparing models at the level of actions, we are considering smaller sample sizes than in the main results, which compare models at the level of easy/medium/hard splits. There is therefore lower statistical power to identify significant result differences, so the results are less certain. We elected not to compare model performance at the level of action differences, because here the sample sizes are very small, so any correlations would not meet significance thresholds.

## G.2  DETAILED DIFFERENCE ANALYSIS

In Section 6.3, we discuss an analysis of the accuracy at the difference level. The vary long table 14 gives the per-difference accuracies and p-values compared for the accuracy against a random guessing baselines. Each difference is associated with an action key, whose description is in table 13.

Table 11: Action-level scores for each model, and their differences compared to the average model score for that action. The model names are abbreviated and the full model names are GPT-4o, Gemini-1.5-Pro, Claude-3.5-Sonnet, LLaVA-Video-7B, Qwen2-VL-7B

| Split | Action | Action Name | Count | Scores GPT | Gemini | Claude | LLaVA-Vid | Qwen2 | Avg |
|---|---|---|---|---|---|---|---|---|---|
| easy | fitness_0 | Hip circle anticlockwise | 129 | 56.6 | 58.1 | 56.6 | 51.9 | 45.0 | 53.6 |
| easy | fitness_3 | lucky cat | 62 | 53.2 | 58.1 | 43.5 | 58.1 | 45.2 | 51.6 |
| easy | fitness_4 | Squat Knee Raise side view | 43 | 65.1 | 69.8 | 37.2 | 69.8 | 55.8 | 59.5 |
| easy | fitness_6 | Hip circle clockwise | 123 | 58.5 | 76.4 | 69.9 | 56.1 | 52.8 | 62.8 |
| medium | ballsports_0 | basketball jump shot | 96 | 55.2 | 57.3 | 52.1 | 57.3 | 61.5 | 56.7 |
| medium | ballsports_1 | basketball mikan layup | 148 | 56.8 | 49.3 | 46.6 | 51.4 | 56.1 | 52.0 |
| medium | ballsports_2 | basketball reverse layup | 125 | 46.4 | 55.2 | 49.6 | 44.0 | 50.4 | 49.1 |
| medium | ballsports_3 | soccer penalty kick | 70 | 51.4 | 60.0 | 57.1 | 70.0 | 54.3 | 58.6 |
| medium | diving_0 | diving | 240 | 53.8 | 52.1 | 53.3 | 50.8 | 54.2 | 52.8 |
| medium | fitness_1 | Opening and closing step left side first | 186 | 57.5 | 54.8 | 52.7 | 51.1 | 51.6 | 53.5 |
| medium | fitness_2 | car deadlift | 137 | 55.5 | 62.8 | 62.0 | 47.4 | 54.0 | 56.4 |
| medium | fitness_5 | Squat Knee Raise diagonal view | 70 | 35.7 | 32.9 | 38.6 | 62.9 | 44.3 | 42.9 |
| medium | fitness_7 | Opening and closing step right side first | 155 | 52.9 | 52.9 | 63.2 | 49.7 | 52.9 | 54.3 |
| hard | music_0 | piano | 94 | 51.1 | 51.1 | 58.5 | 51.1 | 52.1 | 52.8 |
| hard | music_1 | guitar | 20 | 55.0 | 40.0 | 45.0 | 50.0 | 65.0 | 51.0 |
| hard | surgery_0 | Knot Tying | 237 | 47.7 | 43.5 | 43.5 | 46.4 | 44.3 | 45.1 |
| hard | surgery_1 | Suturing | 309 | 48.9 | 51.5 | 48.2 | 47.6 | 50.2 | 49.3 |
| hard | surgery_2 | Needle Passing | 211 | 51.7 | 46.9 | 49.8 | 48.8 | 50.2 | 49.5 |

Table 12: Action-level difference scores for each model relative to the mean model score on that action. This is the difference with respect to the table 11. The model names are abbreviated and the full model names are GPT-4o, Gemini-1.5-Pro, Claude-3.5-Sonnet, LLaVA-Video-7B, Qwen2-VL-7B

| Split | Action | Action Name | Count | Differences GPT | Gemini | Claude | LLaVA-Vid | Qwen2 |
|---|---|---|---|---|---|---|---|---|
| easy | fitness_0 | Hip circle anticlockwise | 129 | 2.9 | 4.5 | 2.9 | -1.7 | -8.7 |
| easy | fitness_3 | lucky cat | 62 | 1.6 | 6.5 | -8.1 | 6.5 | -6.5 |
| easy | fitness_4 | Squat Knee Raise side view | 43 | 5.6 | 10.2 | -22.3 | 10.2 | -3.7 |
| easy | fitness_6 | Hip circle clockwise | 123 | -4.2 | 13.7 | 7.2 | -6.7 | -9.9 |
| medium | ballsports_0 | basketball jump shot | 96 | -1.5 | 0.6 | -4.6 | 0.6 | 4.8 |
| medium | ballsports_1 | basketball mikan layup | 148 | 4.7 | -2.7 | -5.4 | -0.7 | 4.1 |
| medium | ballsports_2 | basketball reverse layup | 125 | -2.7 | 6.1 | 0.5 | -5.1 | 1.3 |
| medium | ballsports_3 | soccer penalty kick | 70 | -7.1 | 1.4 | -1.4 | 11.4 | -4.3 |
| medium | diving_0 | diving | 240 | 0.9 | -0.8 | 0.5 | -2.0 | 1.3 |
| medium | fitness_1 | Opening and closing step left side first | 186 | 4.0 | 1.3 | -0.9 | -2.5 | -1.9 |
| medium | fitness_2 | car deadlift | 137 | -0.9 | 6.4 | 5.7 | -8.9 | -2.3 |
| medium | fitness_5 | Squat Knee Raise diagonal view | 70 | -7.1 | -10.0 | -4.3 | 20.0 | 1.4 |
| medium | fitness_7 | Opening and closing step right side first | 155 | -1.4 | -1.4 | 8.9 | -4.6 | -1.4 |
| hard | music_0 | piano | 94 | -1.7 | -1.7 | 5.7 | -1.7 | -0.6 |
| hard | music_1 | guitar | 20 | 4.0 | -11.0 | -6.0 | -1.0 | 14.0 |
| hard | surgery_0 | Knot Tying | 237 | 2.6 | -1.6 | -1.6 | 1.4 | -0.8 |
| hard | surgery_1 | Suturing | 309 | -0.4 | 2.2 | -1.0 | -1.7 | 0.9 |
| hard | surgery_2 | Needle Passing | 211 | 2.2 | -2.6 | 0.3 | -0.7 | 0.8 |

Table 13: Actions keys and their descriptions

| Action | Action description |
| --- | --- |
| ballsports_0 | a person is doing a basketball mid-range jump shot, starting with the ball in their hand, no defense, practice only |
| ballsports_1 | a person does the Basketball drill called the Mikan layup where they start under the basket, do a layup with the right hand and catch it, do a left hand layup and catch it, no defense, practice only |
| ballsports_2 | a person playing basketball does a reverse layup starting from the left side of the basket and lays it up with their right hand on the right hand side, no defense, practice only |
| ballsports_3 | a person does a soccer drill where they do a single penalty kick, practice only, no defense, no goalie |
| diving_0 | competitive diving from 10m |
| fitness_0 | fitness exercise called standing hip circle with hands on hips, one rotation anti-clockwise |
| fitness_1 | fitness exercise called opening and closing step on left side and then opening and closing step on right side |
| fitness_2 | a single free weight deadlift without any weight |
| fitness_3 | fitness exercise called two arm standing lucky cat starting with arms up, one repetition |
| fitness_4 | squat without weights, then knee raise on left side |
| fitness_5 | a squat then a knee raise on left side |
| fitness_6 | fitness exercise called standing hip circle with hands on hips, one rotation clockwise |
| fitness_7 | fitness exercise called opening and closing step on right side and then opening and closing step on left side |
| music_0 | a person is playing scales on the piano |
| music_1 | a person is playing scales on the guitar |
| surgery_0 | The subject picks up one end of a suture tied to a flexible tube attached at its ends to the surface of the bench-top model, and ties a single loop knot. |
| surgery_1 | The subject picks up needle, proceeds to the incision (designated as a vertical line on the bench-top model), and passes the needle through the fake tissue, entering at the dot marked on one side of the incision and exiting at the corresponding dot marked on the other side of the incision. After the first needle pass, the subject extracts the needle out of the tissue, passes it to the right hand and repeats the needle pass three more times. |
| surgery_2 | The subject picks up the needle (in some cases not captured in the video) and passes it through four small metal hoops from right to left. The hoops are attached at a small height above the surface of the bench-top model. |

Table 14: Difference-level accuracy scores for VidDiff. The 'action' values can be looked up at table 13. The grayed columns indicate a p-value < 0.05 for the two-tailed binomial significance test

| Split | Action | Difference Description | Mean score | Num Samples | p-value |
|---|---|---|---|---|---|
| easy | fitness_6 | the head remains more vertical during the rotation | 1 | 13 | 0 |
| medium | fitness_2 | the gaze is forward at the bottom of the deadlift | 1 | 7 | 0.016 |
| medium | ballsports_2 | they gather the ball with both hands | 1 | 2 | 0.5 |
| hard | music_1 | The player uses a plectrum. | 1 | 4 | 0.125 |
| medium | fitness_7 | the motion is faster | 0.93 | 14 | 0.011 |
| medium | ballsports_1 | uses the non-shooting hand (the guide hand) for stabilizing the ball during the shot in the 1st shot | 0.9 | 10 | 0.02 |
| medium | fitness_2 | the knees bend less at the bottom of the deadlift | 0.89 | 19 | 0.004 |
| easy | fitness_6 | the toes are more pointed out | 0.88 | 16 | 0.004 |
| medium | ballsports_1 | they jump higher on the first shot | 0.83 | 12 | 0.107 |
| easy | fitness_6 | the feet stance is wider | 0.83 | 24 | 0.005 |
| medium | fitness_2 | the feet stance is wider | 0.8 | 15 | 0.028 |
| easy | fitness_3 | the feet stance is wider | 0.8 | 5 | 0.313 |
| easy | fitness_0 | the feet stance is wider | 0.8 | 25 | 0.003 |
| medium | ballsports_1 | uses the non-shooting hand (the guide hand) for stabilizing the ball during the shot in the 2nd shot | 0.8 | 10 | 0.088 |
| medium | ballsports_0 | the shooter's arm is more extended towards the basket | 0.79 | 14 | 0.122 |
| medium | fitness_7 | the arms are elevated in an uneven way on the first step | 0.75 | 20 | 0.03 |
| hard | surgery_1 | The left graser supports the right grasper, by pressing down on the tissue. | 0.75 | 4 | 0.5 |
| medium | ballsports_0 | as the shooter begins extending the elbow to shoot, the non-shooting hand (the guide hand) is on the side of the ball and does not influence the balls trajectory | 0.75 | 12 | 0.107 |
| medium | ballsports_3 | they kick the ball harder | 0.75 | 12 | 0.107 |
| medium | ballsports_2 | the non-jumping leg has a more elevated knee | 0.73 | 15 | 0.183 |
| medium | fitness_2 | the gaze is forward at the start of the motion | 0.73 | 11 | 0.322 |
| hard | surgery_2 | The instrument tips are never out of view (occluded by instruments, or out of frame) | 0.73 | 11 | 0.322 |
| medium | fitness_7 | the arms reach higher on the first step | 0.71 | 17 | 0.189 |
| hard | surgery_2 | The second grasper is used to stabalize the target. | 0.69 | 26 | 0.093 |
| medium | fitness_2 | the hands are lower at the bottom of the deadlift | 0.68 | 19 | 0.192 |
| easy | fitness_3 | the upper arms are parallel to the ground at the start of the motion | 0.68 | 19 | 0.192 |
| medium | fitness_1 | the torso moves out further closer to the foot during the second step out | 0.67 | 21 | 0.194 |
| easy | fitness_6 | the range of motion in the hips is larger | 0.67 | 24 | 0.156 |
| easy | fitness_6 | The upper body rocks more in a forward-backward way | 0.67 | 9 | 0.492 |
| medium | fitness_2 | the body is more locked out at the top of the deadlift | 0.67 | 6 | 0.625 |
| medium | fitness_1 | the arms reach higher on the first step | 0.67 | 18 | 0.243 |
| medium | ballsports_2 | the ball is closer to the corner of the square on the backboard | 0.67 | 9 | 0.492 |
| easy | fitness_0 | The upper body rocks more in a forward-backward way | 0.67 | 18 | 0.243 |
| medium | ballsports_3 | they rotate their hips more during the strike | 0.67 | 18 | 0.243 |
| medium | ballsports_3 | the head is facing the ball leading up to the strike | 0.67 | 6 | 0.625 |
| hard | music_0 | Rhythmic consistency is better maintained in Video A than in Video B. | 0.67 | 12 | 0.387 |
| easy | fitness_0 | the toes are more pointed out | 0.65 | 17 | 0.297 |
| medium | diving_0 | The size and volume of the splash created upon entry is larger for video A than video B. | 0.65 | 48 | 0.052 |
| easy | fitness_6 | the speed of hip rotation is faster | 0.64 | 25 | 0.122 |
| medium | ballsports_1 | they jump with two feet on the 1st shot | 0.64 | 11 | 0.451 |
| medium | diving_0 | Diver enters the water at an angle closer to 90 degrees in video A than in video B. | 0.63 | 30 | 0.161 |
| medium | fitness_1 | the arms are elevated in an uneven way on the second step | 0.63 | 19 | 0.192 |
| easy | fitness_0 | the range of motion in the hips is larger | 0.63 | 19 | 0.192 |
| hard | surgery_0 | The tube in Video A moves more than in Video B. | 0.63 | 27 | 0.194 |
| easy | fitness_4 | the squat is deeper, measured by angle of the thigh to the ground | 0.63 | 16 | 0.244 |
| easy | fitness_3 | the toes are more pointed out | 0.63 | 8 | 0.438 |
| medium | fitness_7 | the toes are more pointed outwards on the first step out | 0.63 | 8 | 0.438 |
| hard | music_0 | Forearm movement is more controlled and minimal in Video A than in Video B. | 0.62 | 13 | 0.419 |
| medium | fitness_1 | the second step out is wider | 0.62 | 13 | 0.419 |
| medium | fitness_1 | the arms reach higher on the second step | 0.61 | 18 | 0.334 |
| easy | fitness_0 | the head remains more vertical during the rotation | 0.61 | 18 | 0.334 |
| medium | ballsports_1 | the non-jumping leg has a more elevated knee in the 2nd shot | 0.6 | 5 | 0.625 |
| medium | ballsports_0 | the shooter's jump is more vertical than forward | 0.6 | 5 | 0.625 |
| medium | ballsports_2 | the gaze is more up and forward instead of down on the 2nd last step | 0.6 | 10 | 0.41 |
| medium | ballsports_1 | the arm is more fully extended towards the basket in the follow through in the 2nd shot | 0.6 | 15 | 0.305 |
| medium | ballsports_0 | the shooter's feet stance is wider when starting the shooting motion | 0.6 | 10 | 0.41 |
| medium | ballsports_1 | the body moves more forward during the 2nd shot, rather than up or back | 0.6 | 10 | 0.41 |
| medium | ballsports_3 | the non-kicking foot is planted closer to the ball | 0.6 | 10 | 0.41 |
| hard | surgery_0 | The tension on the suturing material and the tissue is better controlled in Video A than in Video B. | 0.59 | 32 | 0.162 |
| easy | fitness_6 | the hand position is higher on the body | 0.58 | 12 | 0.451 |
| medium | ballsports_3 | the non-kicking foot is planted more next to the ball and less behind the ball | 0.58 | 12 | 0.451 |
| medium | ballsports_1 | they catch the ball in a higher position on the 1st shot | 0.58 | 12 | 0.451 |
| easy | fitness_0 | the hand position is higher on the body | 0.58 | 12 | 0.451 |
| hard | music_0 | The speed of playing is higher in Video A than in Video B. | 0.58 | 19 | 0.352 |
| hard | surgery_1 | The instrument tips are never out of view (occluded by instruments, or out of frame) | 0.58 | 19 | 0.352 |
| hard | surgery_1 | The suturing speed is higher in Video A than in Video B | 0.58 | 52 | 0.157 |
| hard | surgery_2 | The movement of the needle through the hoop is more radial in Video A than in Video B. | 0.58 | 26 | 0.288 |
| hard | surgery_1 | The tension on the suturing thread is lower in Video A than in Video B. | 0.57 | 28 | 0.279 |
| medium | fitness_7 | the arms are elevated in an uneven way on the second step | 0.56 | 16 | 0.349 |
| medium | fitness_1 | the arms are elevated in an uneven way on the first step | 0.56 | 16 | 0.349 |
| hard | surgery_2 | The number of movements to arrange the needle before threading is lower in Video A than in Video B. | 0.56 | 41 | 0.223 |
| hard | surgery_1 | The movement is more fluid in Video A than in Video B. | 0.56 | 43 | 0.218 |
| hard | surgery_1 | The grasper in Video A is more quickly positioned on the needle than in Video B. | 0.56 | 43 | 0.218 |
| medium | fitness_7 | the toes are more pointed outwards on the second step out | 0.56 | 9 | 0.492 |
| medium | diving_0 | Diver rotates forward relative to themselves. | 0.56 | 27 | 0.299 |
| medium | ballsports_2 | they get deeper knee bend before jumping | 0.56 | 9 | 0.492 |
| easy | fitness_0 | the speed of hip rotation is faster | 0.55 | 20 | 0.32 |
| medium | ballsports_1 | they jump higher on the second shot | 0.55 | 11 | 0.451 |
| easy | fitness_4 | the toes are more pointed out | 0.54 | 13 | 0.419 |
| medium | ballsports_0 | the shooter's feet position are more staggered, meaning the feet are at a different distance from the basket | 0.54 | 13 | 0.419 |

| | | | | | |
|---|---|---|---|---|---|
| medium | diving_0 | Duration from jump off the board to water entry in longer in video A than in video B. | 0.54 | 39 | 0.251 |
| medium | ballsports_0 | the shooter's knees are more bent before taking the shot | 0.54 | 13 | 0.419 |
| medium | ballsports_2 | they use the non-shooting hand (the guide hand) for stabilizing the ball during the shot | 0.53 | 17 | 0.371 |
| medium | fitness_5 | the squat is deeper, measured by angle of the thigh to the ground | 0.53 | 17 | 0.371 |
| hard | surgery_2 | The needle is grasped closer to the tip in Video A than in video B. | 0.52 | 21 | 0.336 |
| hard | surgery_2 | The force on the target is lower in Video A than in Video B | 0.51 | 37 | 0.257 |
| medium | diving_0 | Diver's body is more straight in video A than in video B. | 0.51 | 39 | 0.251 |
| medium | ballsports_1 | they have better balance when landing on the 1st shot | 0.5 | 2 | 1 |
| medium | diving_0 | Speed at which divers rotate during the dive in larger in video A than in video B. | 0.5 | 38 | 0.257 |
| medium | fitness_7 | the arms reach higher on the second step | 0.5 | 14 | 0.419 |
| medium | ballsports_1 | they have better balance when landing on the 2nd shot | 0.5 | 2 | 1 |
| hard | music_1 | Smooth transitions between strings with minimal disruption to the rhythm or tempo. Transitions in Video A are smoother than in Video B. | 0.5 | 2 | 1 |
| hard | music_1 | Guiatrist uses finger vibrato. | 0.5 | 2 | 1 |
| medium | ballsports_3 | the body (or torso) is more facing the net, or more 'square' to the net | 0.5 | 12 | 0.451 |
| medium | fitness_7 | the first step out is wider | 0.47 | 17 | 0.297 |
| hard | surgery_0 | The movements in Video A are more precise than in Video B. | 0.47 | 34 | 0.216 |
| easy | fitness_3 | the speed of the arms is faster | 0.47 | 17 | 0.297 |
| medium | ballsports_2 | they land on two feet | 0.47 | 17 | 0.297 |
| medium | ballsports_2 | they follow through more and towards the basket | 0.47 | 15 | 0.305 |
| medium | fitness_2 | the toes are more pointed out | 0.47 | 15 | 0.305 |
| medium | fitness_2 | there is a pause at the bottom of the deadlift | 0.47 | 15 | 0.305 |
| medium | fitness_7 | the second step out is wider | 0.46 | 13 | 0.314 |
| hard | surgery_1 | The needle is inserted in the fabric more perpendicular to the incision. | 0.46 | 13 | 0.314 |
| medium | fitness_1 | the toes are more pointed outwards on the first step out | 0.46 | 13 | 0.314 |
| hard | music_0 | The smoothness of thumb crossing is more evident in Video A than in Video B. | 0.46 | 13 | 0.314 |
| easy | fitness_3 | the upper arms more stable through the entire motion | 0.46 | 13 | 0.314 |
| medium | ballsports_1 | they jump off the right foot for right-hand shot (the 2nd shot) | 0.45 | 11 | 0.322 |
| medium | ballsports_2 | before raising the ball to shoot, the ball is more to the right side of the hip | 0.45 | 11 | 0.322 |
| hard | surgery_1 | The force is applied in a more radial way in Video A than in Video B. | 0.45 | 31 | 0.192 |
| medium | fitness_7 | the torso moves out further closer to the foot during the first step out | 0.44 | 18 | 0.243 |
| medium | fitness_2 | the arms are in front of the body at the bottom of the deadlift | 0.44 | 9 | 0.328 |
| hard | surgery_2 | The passage of the needle between two hands is more fluid in Video A than in Video B. | 0.44 | 25 | 0.266 |
| medium | fitness_5 | the feet stance is wider | 0.44 | 16 | 0.349 |
| hard | music_0 | The wrist should be straight and not dipped or raised, facilitating fluid motion and avoiding strain. The wrist position is more appropriate in Video A than in Video B. | 0.43 | 14 | 0.244 |
| medium | fitness_2 | the entire motion is faster | 0.43 | 21 | 0.194 |
| hard | surgery_0 | The movements in Videos A are faster than in Video B. | 0.43 | 42 | 0.116 |
| hard | surgery_0 | More errors are corrected in Video A than in Video B. | 0.42 | 31 | 0.131 |
| hard | surgery_2 | The thread is more efficiently managed in Video A than in Video B. | 0.42 | 24 | 0.156 |
| medium | fitness_1 | the toes are more pointed outwards on the second step out | 0.41 | 17 | 0.189 |
| hard | surgery_1 | The instrument applies more force to the tissue and needle in Video A than in Video B | 0.41 | 44 | 0.078 |
| hard | surgery_0 | The movements in Video A are more efficient than in Video B. | 0.4 | 42 | 0.076 |
| medium | ballsports_1 | the body moves more forward during the 1st shot, rather than up or back | 0.4 | 5 | 0.625 |
| medium | ballsports_0 | the shooter's feet are oriented more square to the basket when starting the shooting motion, meaning the feet point more forward | 0.38 | 13 | 0.175 |
| medium | ballsports_0 | as the shooter begins extending the elbow to shoot, the ball is more in front of the body, rather than behind the head | 0.38 | 16 | 0.244 |
| hard | surgery_1 | The dot is more accurately hit in Video A than in Video B | 0.36 | 14 | 0.122 |
| hard | music_0 | The body is closer to the piano in Video A than in Video B. | 0.35 | 17 | 0.094 |
| medium | fitness_5 | the speed of the whole motion is faster | 0.35 | 17 | 0.094 |
| medium | ballsports_2 | they release the ball at a higher position | 0.35 | 20 | 0.148 |
| medium | ballsports_1 | the non-jumping leg has a more elevated knee in the 1st shot | 0.33 | 9 | 0.141 |
| hard | surgery_0 | Both graspers are used efficiently. | 0.33 | 9 | 0.141 |
| hard | surgery_0 | The surgeon in Video A stops more often to plan next steps than the surgeon in Video B. | 0.33 | 3 | 0.25 |
| hard | music_0 | There are more wrong note corrections in Video A than in Video B. | 0.33 | 6 | 0.188 |
| medium | fitness_1 | the motion is faster | 0.33 | 18 | 0.065 |
| medium | ballsports_1 | they have more fluid motion in moving between the shots | 0.33 | 12 | 0.107 |
| hard | music_1 | The left fingers in Video A are more curved / less collapsed than in video B. | 0.33 | 3 | 0.25 |
| medium | fitness_7 | the torso moves out further closer to the foot during the second step out | 0.33 | 9 | 0.141 |
| medium | diving_0 | Diver faces the water at jump off. | 0.32 | 19 | 0.044 |
| medium | fitness_1 | the first step out is wider | 0.31 | 16 | 0.133 |
| medium | fitness_5 | during the squat descent, the knees cave inwards, instead of tracking over the feet | 0.3 | 20 | 0.074 |
| medium | fitness_1 | the torso moves out further closer to the foot during the first step out | 0.29 | 17 | 0.036 |
| hard | surgery_1 | The grasper grasps the needle approximately 2/3 from the needle tip. The needle is grasped more precisely in Video A than in Video B. | 0.29 | 34 | 0.006 |
| easy | fitness_4 | the feet stance is wider | 0.29 | 14 | 0.044 |
| hard | music_1 | The unused left finger tips in Video A stay closer to the board than in video B. | 0.25 | 4 | 0.5 |
| hard | surgery_0 | The suturing thread tangles. | 0.24 | 17 | 0.01 |
| medium | ballsports_1 | the arm is more fully extended towards the basket in the follow through in the 1st shot | 0.18 | 11 | 0.011 |
| hard | music_1 | Fingers should press strings at the center of the frets, avoiding the metal fret bars for clear sound production. Video A shows more accurate finger placement on the fretboard than Video B. | 0 | 2 | 0.5 |
| hard | music_1 | Only one finger of the left hands rests on a string at a time. | 0 | 3 | 0.25 |

Table 15: Evaluating 'easy' split with variable video fps for three models. Our evaluation protocol chooses 4fps.

|  | 1 fps | 2 fps | 4 fps | 8 fps | average |
|---|---|---|---|---|---|
| **GPT-4o** | 58.0 | 59.4 | 58.8 | 59.10 | 58.8 |
| **Gemini-1.5-Pro** | 59.7 | 66.9 | 65.8 | 66.9 | 64.8 |
| **Claude-3.5-Sonnet** | 58.1 | 58.5 | 56.6 | 52.9 | 56.5 |

### G.3  RESULTS: DEPENDENCE ON FPS

The frame sampling rate, fps, is an important consideration for evaluating fine-grained actions. While typical video benchmarks like Video-MME Fu et al. (2024) sample videos at 1fps, we have sampled at a higher rate depending on category. The categories with shorter videos were sampled at a higher rate: 4fps for 'fitness', 5fps for 'ballsports', and 6fps for 'diving' (they are slightly different so they can be compatible with fps in the source dataset). We chose this relatively higher rate because we are interested in more fine-grained differences, while prior benchmarks are more coarse-grained; however we did not sample at even higher due to practical cost constraints of processing too many frames. The longer videos 'surgery' and 'music' were sampled at 1fps: these are longer videos where differences are discernible at lower sampling rates, and where the longer videos make high-fps sampling impractical.

To show that our fps is reasonable, we tested the three closed-source models on a range of fps levels on the 'easy' subset of closed evaluation. We chose this set because this is where statistically significant differences were clear. The results are in table 15.

Across all models, the sampling rate that we use, 4 fps, has reasonable scores, either at or above the average over the other fps values. For all models, the variability is low: GPT's scores are within 0.8 points of the average; all other models have scores within 2.1 points of the average (except for the low sampling rate of 1fps in Gemini, where it degrades by 5.2 points). Moreover, the optimal fps is different for different models.

To help explain the results, we refer to the qualitative examples in the main results sections. The only 'success cases' for all our models were those having easy localization, and coarse differences. We hypothesize that fps is not important for these cases. Where fps is likely important -– fine-grained multiframe reasoning – the current LMMs cannot perform better than random. So although 2fps currently has good performance, we believe that as LMMs improve, they will perform better on subtle motions and using a higher fps will be important.

### G.4  RESULTS: QWEN2-VL OPEN EVALUATION

Qwen2-VL performs especially poorly in open evaluation, which we investigate here. The key issue is that Qwen2-VL-7b was failing to follow the evaluation prompt, while the other compared models did follow it. We sampled 3 video pairs for each action and manually inspected Qwen's responses, identifying multiple key issues. Below, we list each issue, and provide a quantitative estimate for the prevalence of each issue.

- (45% of differences) Proposing differences not relevant to *how* to perform actions, but instead are visual things like "The person in video a is wearing a blue jacket, while the person in video b is wearing a plaid shirt." We estimated prevalence by using a gpt-4o query that we manually prompt engineered.

- (26% of differences) Proposing a difference that is actually not a difference, e.g. "The person in video a is performing the exercise with their arms out to the sides, while the person in video b is performing the exercise with their arms out to the sides." We estimated prevalence by using a gpt-4o query that we manually prompt engineered.

- (56% of differences) are repeated, meaning when trying to propose multiple differences, it proposes the same difference multiple times. We could directly measure this prevalence exactly.

- (23% of actions) Proposing only a small number of differences – less then half as many as what is prompted for. We could directly measure this prevalence exactly.
- ($\leq$5% of differences) Proposing vague differences that are harder to interpret visually like "The player in video a has a more versatile and adaptable skill set than the player in video b". We estimated prevalence by using a gpt-4o query that we manually prompt engineered.

Overall, only 31.9% of proposed differences by Qwen did not suffer from any of these errors. (Note that some differences suffered from multiple errors at the same time)

### G.5 NO MULTIPLE CHOICE BIAS IN CLOSED EVALUATION

In multiple choice benchmarking, models may be biased towards one particular option, which can impact evaluation robustness. We find no evidence of this. Firstly, in closed evaluation, the A/B ratio is 0.493/0.507. Second, we test the impact of video order on GPT-4o for the 'fitness' category, which has samples in the easy and medium subsets(sample size 193). We test flipping the order of videos which flips the A/B answer. The performance is 54.8% in the original evaluation, and reversing the order of videos gives performance of 55.5%, showing a 0.7% difference. This result suggests that the performance on VidDiffBench is not significantly sensitive to video order.

### G.6 EXPERIMENT ON DUPLICATING VIDEO

One idea to validate the reasonable-ness of the benchmark is to check what happens when passing an identical video as A and B to the system – we should expect that in closed evaluation, the predictions should be A/B 50% of the time. We did this experiment on the closed setting, for the 'easy' subset for GPT-4o. Over two random seeds, the results were 49.3 and 50.2. This is an interesting validation check that the benchmark passes.

### G.7 NO MULTIPLE CHOICE BIAS IN CLOSED EVALUATION

In multiple choice benchmarking, models may be biased towards one particular option, which can impact evaluation robustness. We find no evidence of this. Firstly, in closed evaluation, the A/B ratio is 0.493/0.507. Second, we test the impact of video order on GPT-4o for the 'fitness' category, which has samples in the easy and medium subsets(sample size 193). We test flipping the order of videos which flips the A/B answer. The performance is 54.8% in the original evaluation, and reversing the order of videos gives performance of 55.5%, showing a 0.7% difference. This result suggests that the performance on VidDiffBench is not significantly sensitive to video order.

### G.8 SAMPLE RETRIEVALS

In our methods, we leverage a frame localizer to find either a single frame or a small sequence of frames. These localized frames are then passed to the next stage. In fig. 4, we show a few examples of predictions vs ground truth for some frames

## H VIDDIFF METHOD

The code for VidDiff method is available at this GitHub repo https://github.com/jmhb0/viddiff.

Here we show the prompts used in the different components.

**Proposer stage** Part 1 chooses candidate differences (Open setting only):

```
I have two videos of an action with the following description: "{action}".

Propose a set of 'differences' in how this action may be performed between the two
    videos.
For each difference, give a 'name', 'description', 'query_string', and 'num_frames'.

The 'name' is a very short name that can be used as a key.
```

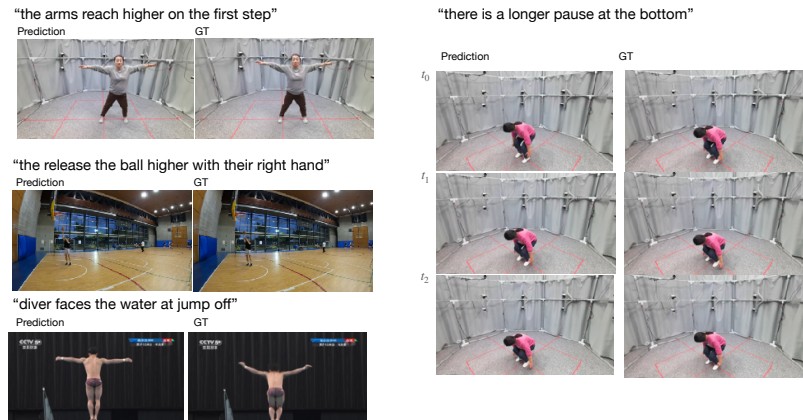

Figure 4: Sample frame localizations: prediction vs ground truth.

```
The 'description' is a slightly longer description of the difference.
The 'query_string' is the same as 'description'.
The  descriptions should be visual and about how the action is performed. For example '
    the jump is higher', or 'the arm is more straight'.
The difference descriptions should not refer to a specific video. For example you would
    not say 'the jump in video B is higher'.
Instead, the 'description' would be 'the jump is higher', and the statemet could be more
     true of one video or the other.

Now suppose you had to judge whether the difference was stronger in one video, but you
    could only look at individual frames.
 - What's the smallest number of frames you need, assuming the frame is well-chosen? Put
     the answer in 'num_frames'. The answer should be '1' or 'gt_1' (meaning 'greater
    than 1').
 - Once you have the frames to compare, create a 'query_string' that is a simple
    statement about this difference that is more true in one video vs the other video
     (based on the frames only). For example "the left leg is higher" or "movement is
    faster".

List {n_differences} differences.
Return a json like this, where the differences keys are stringified ints starting from
    0.
{
 '0' : {
  "name" : "...",
  "description" : "...",
  "query_string" : "...",
  "num_frames": "1|gt_1",
 },
 ...
}
```

Proposer part 2 estimates sub-action stages:

```
I have two videos of an action with the following description: "{action}".

Provide a 'stage transcript' as a list. These are sub-actions that make up that action.
Give 5 steps or fewer in the action transcript.

For each stage, give a 'name' for the stage, and a 'description' of that stage.

For each stage, give a list of 'retrieval_strings'.
```

```
These are strings that describe what is visible in the frame.
Only describe the visual features. Only describe what is visible in a single frame.
    Focus on appearance. Focus on pose. Do not use the name of the action. Start each
    string with something similar to "A photo of a ...".
Give at least {n_retrieval_keys} retrieval strings per stage.

Return a json like this:
{
 "stages" : [
  {
  "name : "",
  "description" : "...",
  "retrieval_strings" : ["A photo of a ...", ...],
  },
  ...
]}
```

And proposer part 3 does linking between those stages

```
I have two videos of an action with the following description: "{action}".

Here are a list of stages or subactions that make up that action:
{stages}

We also have differences in how this action may be performed between the two videos.
The differences are specific statements that are more true in one video vs another.
Here they are:
{differences}

Now we need to match each differences to a stage.
Return a list of the stages using their names.
If a difference is relevant to a particular stage, put its name in the 'difference' list
    .
It's okay for a 'difference' to be visible in multiple stages.
It's okay for some stages to have no difference.
Refer to stages and differences by their 'name' attribute.

Return a json like this:
[
 "<stage_name0>" : ["<difference_name0>", "<difference_name1>", ...],
 "<stage_name1>" : [],
 "<stage_name2>" : ["<difference_name1>", "<difference_name2>", ...],
 ...
]
Please be careful. Every difference must appear at least once
```

**Frame Differencer**   The prompt also takes the image frames.

```
I have two videos of people performing an action with description: "{action}".
The first {num_frames} frames are from video A and the last {num_frames} frames are from
    video B.
For each video, the frames are very close together in the video: they are {time_diff}
    seconds apart.
Which one shows more of the variation with this description: "{query_string}"?
(a) video 1, (b) video 2, (c) similar or can't tell.
Answer in json:  {'answer_detailed' : "...", 'answer':"a|b|c"}""",
```

# I   COMPUTATIONAL COSTS FOR VIDDIFF METHOD

Our method's runtime is less than one minute per video pair using an A6000 GPU for running CLIP inference Radford et al. (2021). Additionally, we utilize the GPT API at an average cost of $0.2

per sample Achiam et al. (2023). Notably, over 95% of the GPT cost arises from verbose VLM responses. Attempts to prompt for shorter responses resulted in degraded performance.

Methodologically, we rely on pre-trained zero-shot models, which limits their applicability in specialized domains, as discussed in our results. For evaluation, the Open setting formulation necessitates an LLM in the evaluation pipeline. One challenge is the subjectivity in annotations: determining which differences are relevant and what magnitude of difference is significant, though we thoroughly discuss this problem and mitigations.

