# OpenReview forum: "Video Action Differencing"
_ICLR.cc/2025/Conference — ICLR 2025 Poster_

### Official Review · Reviewer_8a8a · 2024-10-18

**Soundness:** 3
**Presentation:** 4
**Contribution:** 3
**Rating:** 6
**Confidence:** 4

**Summary:**

This paper introduces the first large-scale video action differencing dataset, presenting a novel task of identifying differences between videos depicting the same action. The authors compile over 500 video pairs from existing datasets across five categories: Fitness, Ball Sports, Diving, Music, and Surgery. These videos are then assigned to annotators along with 147 distinct descriptions. Annotators must indicate which video (A or B) most closely aligns with each description. For example, given two videos of different actors performing a squat, a description might read "deeper squat," and the annotator would select A or B based on which video demonstrates the deeper squat. To ensure dataset quality, 25% of the initial annotations undergo re-annotation, revealing a very low discrepancy rate. The dataset also includes action localization (pinpointing where the action occurs in the video) and specific key points for each action (e.g., when knees start to bend).

The authors also develop an agentic model called VidDiff to address the action differencing challenge. VidDiff employs several Large Language Models (LLMs) and Vision Language Models (VLMs) as agents to solve specific aspects of the problem: proposing potential differences based on the action description, localizing frames where such actions might occur, and finally specifying which video (A or B) corresponds to the observed difference. VidDiff outperforms other zero-shot VLMs in this task.

Lastly, the authors provide ablation experiments that highlight the challenges presented by their new benchmark.

**Strengths:**

### Originality
- **A novel task**: This paper introduces the new task of video action differencing with natural language. While related tasks, such as difference captioning, have been explored to provide a coarse comparison between videos, no prior work has tackled video action differencing in the same way—focusing on fine-grained differences described in natural language.
- **A challenging benchmark**: The proposed benchmark, VidDiffBench, is comprehensive, covering five categories of instructional videos. It has proven to be highly challenging, even for top-performing closed-source vision-language models (VLMs).
- **An agent-based system**: The paper presents an agent-based system that decomposes the task, achieving better performance than existing VLMs.
### Clarity
The flow of ideas is straightforward, making the paper easy to follow and understand.

### Significance
The paper convincingly demonstrates the importance of video action differencing, and the introduction of the new benchmark is likely to inspire further research in this area.

**Weaknesses:**

### Unproven claims
- In the introduction, the authors claim they will address the challenges of *precise temporal alignment and the need for fine-grained understanding of action dynamics*. However, it remains unclear how they specifically solve the issue of temporal alignment. Could you elaborate on how you solve this issue or point us to the location where it is addressed?

### Benchmark and results
- Similar datasets are presented in the related work section; however, since this work is primarily a benchmark paper, more comparisons with existing benchmarks would be make the differences clearer (e.g., similar to Table 1 but with other datasets in the first column). Consider adding what is unique about each dataset and how the current dataset differs.
- As a benchmark paper, we would expect more results from other open-source VLMs (especially those addressing video data such as LLaVA-video) to better understand their limitations and make it easier for other researchers to work with this benchmark.

### Clarity
- 557 or 656 video pairs? In the abstract, the authors state that the dataset contains *557 video pairs...  4,719 fine-grained action differences* (line 013-014), but on line 260, they mention *656 video pairs, 5,580 annotated differences*. Clarification needed on which is correct.
- Figure 1: The distinction between the first and second row is unclear, yet the caption claims these represent two different challenges. These two challenges are not discussed elsewhere in the paper and don't seem to be related to the dataset splits. Please clarify this.

**Questions:**

See weaknesses, plus the following:

- Which LLMs/VLMs are used for the *Difference Proposer* and *Action Differencer*?
- How does the benchmark handle cases of inverse correlation? For example, would *lower squat in video A* be equivalent to *higher squat in video B*?
- Since the videos are not curated, factors such as different camera angles, varying FPS, or differences in the actor's height could introduce biases in the annotations and results. How do the authors address these potential biases?

---

> ### Author Response · Authors · 2024-11-24
> **Comment 1/2 for reviewer 8a8a**
>
> We’d like to thank reviewer 8a8a for recognising the strength of the work, especially the significance of the new proposed task of Video Action Differencing, and the originality of all three of our contributions – the task, benchmark, and method.  We now hope to address each of the raised concerns:
>
> ## Addressing temporal alignment
> Consider the example in Fig.6 row, 2, left, the action is basketball layup, and the difference is “Non-shooting hand guides the ball”. A human would identify the segment in the video where the person is about to release the ball in both videos, and then compare these segments. This is the challenges that we talk about: differencing first requires aligning the sub-actions in the two videos, and after that step, the visual comparison can be done on the two aligned segments. To solve this, our approach performs alignment in a similar way: the ‘frame localizer’ does temporal segmentation in each video, and then the localized segments are both passed to the ‘action differencer’ to compare the visual segments. Having retrieved and aligned the frames, the ‘action differencer’ can work with a smaller segment of video – possibly only a single pair of frames. We have updated the text in our method section to more explicitly say how we have solved the key problem.
>
> ## Related work
> We have added an extended discussion in the appendix about “video comparison datasets”, and summarized it briefly in the main related works section. The high level takeaway is that no other dataset has labels for fine-grained comparison while having a large scale. One relevant prior datasets considers very coarse-grained differences in instructional videos, for example identifying that a different ingredient was used in a cooking recipe (Nagarajan & Torresani 2024). Other works do consider more fine-grained differences, but they either annotate only a single binary variable like “which is more skilled” (EPIC-Skills2018 by Doughty et al 2018), or do not have any annotations (Balakrishnan et al, 2015); both of these dataset examples are small with fewer than 100 pairs.
>
> ## Scale of benchmarking
> In our updated version, we have increased the scale of the benchmark. We’ve added the closed-source Claude-3.5-Sonnet, and the most recent open-source video model, LLaVA-Video-7B. We have updated the main table, and for illustrative purposes, here are is the main table on closed evaluation:
>
> |                   | Easy  | Med   | Hard  | Avg   |
> |-------------------|-------|-------|-------|-------|
> | GPT-4o            | 58.8% | 53.0% | 50.1% | 54.0% |
> | Gemini-1.5-Pro    | 65.8% | 51.9% | 49.8% | 55.8% |
> | Claude-3.5-Sonnet | 56.6% | 53.5% | 48.3% | 52.8% |
> | LLaVA-Video       | 56.6% | 52.0% | 48.3% | 52.3% |
> | Qwen2-VL-7B       | 49.0% | 52.6% | 49.6% | 50.4% |
> | VidDiff (ours)    | 65.3% | 55.4% | 50.4% | 57.0% |
>
> Among closed models, Claude performs worse overall than GPT and Gemini. Among open source models, LLaVA-Video is stronger than Qwen2-VL, becoming the only open source model to achieve statistical significance on the easy split.

---

> ### Author Response · Authors · 2024-11-24
> **Comment 2/2 for reviewer 8a8a**
>
> ## Clarity questions
> - The number of videos is 557 – an earlier version of the work had 656, but we removed a set of actions because they only had one annotated difference each. Thank you for noticing that, and we have updated the text.
> - The fig.1 caption refers to two challenges: (i) “identifying alignment between segments” and (ii) “fine-grained image comparison”. This was discussed in the introduction paragraph 2,  “Two critical obstacles are precise temporal alignment and the need for fine-grained understanding of action dynamics”, which we then elaborate. Our qualitative results discussion also identifies these as major challenges. However we agree that the clarity could be greatly improved by using more consistent language about these two challenges throughout the text. As such, we’ve updated the text in the introduction, and the figure caption.
> - LLMs and VLMs in the method] We have updated in the experiment section that the LLM and VLM are both 'gpt-4o-2024-08-06' (state-of-the-art in both at the time of submission), while the embedding model in the localization module is CLIP “ViT-bigG-14” with checkpoint "laion2b_s39b_b160k".
> - [Inverse correlation] You are correct that “squat is lower” and “squat is higher” are equivalent, but where the A/B prediction should be flipped. This is relevant to the open evaluation setting where the model must generate the difference descriptions. Our open evaluation protocol does properly handle this case. First, for the LLM query matching the ground-truth differences to prediction differences – this does match differences that are semantically equivalent but different in sign, and we find many examples in evaluation where this is the case. For each matched difference, we run a second LLM query to check whether the difference string has the opposite meaning, and if it does, then we flip the A/B prediction. Our earlier manuscript version discusses this as an implementation detail in the appendix and shows the prompt (our example of a matching pair was “the arms are more straight” vs “the arms are more bent”). However since this is an important detail that will concern some readers, we have updated the manuscript to discuss this in the task definition and in the open evaluation protocol.
>
>
> ## Variations in the dataset – angles, fps, actor’s height
> Although there is random sampling used in assigning which pairs to compare, we did manually inspect every video in the sampling set. This allowed us to think carefully about the video attributes, including the very important attributes identified here. We will discuss each particular point below, and we’ll also expand on the writing in the Appendix to be more clear about how the videos are curated. In particular, we’ve explained:
> - Camera angles: the change of camera angle perspective does make the task harder. For samples in the ‘Fitness’ category, the camera angle is the same because the source dataset has a fixed camera rig, and we chose to use the same camera angle. For samples in ‘diving’ and ‘surgery’ categories, the camera angle is approximately the same. On the other hand, samples from ‘ballsports’ and ‘music’ categories can change. A related attribute (not mentioned here) is differences in background – similarly the ‘ballsports’ and ‘music’ categories often had different backgrounds as well. Importantly, these attributes were considered when assigning the difficulty splits. This may partly explain why the fitness exercises are all in the easy and medium split.
> - FPS: each video pair has the same fps. In case others want to leverage our code with new videos, our code does handle the case where FPS is different. Specifically, the input config has a value for the target FPS for running inference, and we subsample the video to have this FPS. (If the videos cannot be subsampled to have the exact target fps, then a warning is printed).
> - Impact of different actor heights: this is a very good observation, and we did address it in our annotation instructions. We clarified that all differences on things like distance should be relative to the actor’s height. We gave the example of “wider foot stance”, saying that if a 5ft actor and a 6ft actor both had their legs 3ft apart, then the shorter actor has a “wider foot stance” relative to their height. This reflects what is commonly understood by descriptions like these in skills coaching.
>
> ## References
> Nagarajan & Torresani 2024, “Step Differences in Instructional Video”
>
> Doughty et al 2018, “Who's better? who's best? pairwise deep ranking for skill determination”
>
> Balakrishnan et al, 2015, “Video diff: Highlighting differences between similar actions in videos”

---

> > ### Comment · Reviewer_8a8a · 2024-11-25
> >
> > I appreciate the authors’ thorough answers to my initial concerns. However, I still have reservations that prevent me from raising my score at this time:
> >
> > 1. Performance vs. Complexity: VidDiff’s performance shows only a marginal improvement over GPT-4o, despite utilizing two instances of GPT-4o alongside a localizer. In my view, this imbalance between complexity and performance diminishes the significance of VidDiff as a contribution.
> >
> > 2. Real-World Applicability of VidDiffBench: I remain unconvinced about the practical viability and potential real-world applications of VidDiffBench. It seems to me that the differences it aims to measure would be more effectively and objectively captured using 3D (or even 2D) keypoints.
> >
> > I will maintain my current rating for now and continue monitoring the discussion to decide if an adjustment is warranted.
> >
> > Thank you.

---

> ### Author Response · Authors · 2024-11-26
> **Comment 1/2 for reviewer 8a8a on complexity & real-world applicability**
>
> Thanks for your comments. These are insightful points that line up quite a bit with discussions we’ve had during the project. We discuss them over two posts.
>
> ## Point 1:
> VidDiff is more computationally efficient than the one-stage GPT-4o baseline while maintaining superior performance in the closed setting and comparable results in the open setting. Despite GPT-4o requiring analysis of 40 frames per video—translating to approximately 12,300 tokens—VidDiff processes only 17 localized frames, reducing the token count to about 4,300, a threefold decrease in computational cost (reducing the API cost for evaluating the whole benchmark from approximately `$`18 to `$`6). This efficiency is achieved through an LLM-only Proposer stage and a CLIP-based localization strategy, both of which introduce minimal overhead compared to GPT-4o's visual processing demands.
>
> VidDiff's computational advantages become more pronounced with longer videos, as its cost scales with the number of target differences (processing only 2–6 frames per video pair) rather than linearly with the total frames, as seen in the LMM baseline. Consequently, the efficiency gap widens with longer videos, aligning with recent research exploring efficiency-performance trade-offs, such as Wang et al. (2024).
>
> Our primary contribution is introducing the novel task of Video Action Differencing and developing a comprehensive benchmark to support it. The VidDiff method is a proof-of-concept to demonstrate that the ‘compound’ approach [Zaharia et al., 2024] will work on this problem, and should be explored in future research. The compound approach has two advantages.. First, they will benefit from improvements in zero-shot models for the localization and image understanding. Second, it enables researchers to improve individual stages of the process independently – to facilitate this, VidDiffBench provides stage-wise annotations, giving a robust framework for evaluating performance at each stage.
>
> [Wang et al 2024] “VideoAgent: Long-form Video Understanding with Large Language Model as Agent”
>
> [Zaharia et al 2024] The shift from models to compound ai systems

---

> > ### Author Response · Authors · 2024-11-26
> > **Comment 2/2 for reviewer 8a8a on complexity & real-world applicability**
> >
> > ## Point 2:
> > We advocate for language-based outputs for three reasons: (i) the differences require understanding not just human keypoints, but also how the human relates to objects and the scene; (ii) language is a preferred medium for giving feedback because it is more specific and interpretable than keypoints; and (iii) we can more easily leverage the zero shot foundation models that will continue to improve.
> >
> > ### General scene understanding
> > While all videos involve human actions, much of the feedback requires understanding object interaction or relative position of the human in the scene. All of the ‘surgery’ videos show a tool interacting with a physical toy model. All of the ‘music’ actions involve the hands interacting with an instrument. Sports like soccer have differences like “how close is the foot relative to the ball”. While human keypoints may be important for many actions, the more general Video Action Differencing task requires understanding video more broadly.
> >
> >
> > ### Language feedback
> > Humans often give and receive action performance feedback using language. For example, in the subreddit called ‘FormCheck’ (top 3% of all subreddits by size) users post videos performing exercises like squats and deadlifts and other users provide brief language-only feedback like “keep your shoulders back at the top of the lift”. In computer vision, the EgoExo4D dataset includes “expert commentary”, where a coach narrates a video with targeted feedback like “The dancer's hand is rotated inwardly a bit. Her palm should be facing to the ground”  [Grauman et al, 2024].
> >
> > Consider the action “basketball jump shot”, where an amateur is comparing their action to an expert. They likely have a small number of differences that they should focus on, for example “the arms more extended towards the basket”. For feedback to be effective, it must:
> > Focus on the differences are the most crucial – “not extending the elbow enough” is important, while “shoulders more back” is not important.
> > Identify if the difference is ‘different enough’ –  if the elbow extension is only too little by 2 degrees, then it is not worth highlighting.
> > Focus on which time point the difference matters – they must perceive elbow angle differences only at the point where the person releases the ball, while elbow angle after the shot is taken is not important;this requires temporal understanding.
> >
> > An AI system with natural language feedback can do these things, but keypoints alone cannot. The most naive keypoint approach – providing a visualization of all keypoints, or maybe highlighting their differences – is too hard for the amateur to interpret because it is not specific. The ideal AI system needs to interpret the keypoint information.
> >
> > ### Complementary Role of Keypoint-Based Methods
> > While we emphasize language-based outputs, we acknowledge that methods based on human keypoints or meshes can enhance the Video Action Differencing task. For example, in our staged method, the Frame Differencer could be augmented with keypoint or mesh predictors to improve image comparison quality [Yuan et al., 2022]. Thus, our formulation is potentially complementary to keypoint models – keypoint methods could improve it.
> >
> > ### Zero-shot foundation models
> > Zero-shot foundation models that are based on language will continue to improve with scale, and so our formulation with text output can take advante of that.
> >
> >
> > ## Incorporating your feedback
> > The objections raised here are all very reasonable, and we hope that our discussion is persuasive. If it is, then we will incorporate this discussion about both these points into the writing, especially in the introduction and related work sections.
> >
> > [Yuan et al, 2022] GLAMR: Global Occlusion-Aware Human Mesh Recovery with Dynamic Cameras, CVPR

---

> > > ### Comment · Reviewer_8a8a · 2024-11-26
> > >
> > > Thank you to the authors for thoroughly addressing my additional concerns. I now have no further issues to raise.
> > >
> > > I have updated my ratings for both presentation and contribution and have increased my overall score from 5 to 6.

---

> > > > ### Author Response · Authors · 2024-11-26
> > > >
> > > > We'd like to that the reviewer again for the thoughtful and constructive discussion.

---

### Official Review · Reviewer_dA39 · 2024-11-01

**Soundness:** 2
**Presentation:** 1
**Contribution:** 4
**Rating:** 8
**Confidence:** 5

**Summary:**

In this paper, a new task called video action differencing is proposed which aims for models to be able to understand fine-grained differences between multiple videos of people performing the same action. A new benchmark dataset is collected, named VidDiffBench, which includes 5 categories from 4 different existing datasets. Annotations are collected from pairs of videos with statements given per pair of video based on the action (for example video A includes someone jumping higher than Video B for a lay-up shot). There are two main evaluation protocols for this task, a closed set setting, in which the model must predict A or B for each possible description, and a closed set setting in which the method must generate the description. A new method which combines stages named VidDiff is proposed which outperforms standard LMMs on the dataset.

**Strengths:**

* The new task of Video Action Differencing is an interesting new task for video understanding, forcing models to recognise and understand fine-grained differences between two very similar videos.
* The collected dataset combines four datasets with 5 different categories of video, providing a varied test bed for this new task.
* The proposed method performs well on the dataset, outperforming off the shelf LMMs on the task yet still showcase that there is a lot still to work on in this area for future work.

**Weaknesses:**

# Weaknesses

* There are some missing references for skill determination within the related work [a, b, c, d] as another example of fine-grained differences between videos containing the same action.
* Line 196: It is mentioned here in the text that *"Video pairs are randomly sampled within each dataset to ensure a wide range of comparison difficulty, from simple actions to more advanced tasks requiring fine-grained understanding"* This implies that videos of differing actions are compared against one another.
* Section 3.3.2: There are some missing information about the annotators, regarding skill level, total number, renumeration etc.
* For the closed set, a binary classification setup was used as all candidate difference statements which is mentioned to be unbiased on Line 298. However, has this been checked? If videos are not randomly swapped at inference/training time there could have been a bias towards one video or another.
* The open set evaluation seems like it could be prone to some errors/inconsistencies depending on the LLM chosen and how much it could hallucinate/not understand the task and doesn't represent a potentially sound evaluation protocol.
* It is not clear within the paper as to why an LLM was used to choose the easy/medium/hard splits for each of the actions.
* This paper did not feel like an easy read, whilst the grammar/sentence clarity was good. There was a lot of information that is split across the main paper and the appendix which necessitates jumping between them. The structure of the paper could also be improved, the method details occur within the experiments yet are given as a main contribution within the introduction with only a small amount of space given to explain the model. Another major factor for this is that details of the dataset are given before the task is formally defined, which given this is a new task, makes it harder to read than it should be.

# Additional Comments
Line 158 is referring to the wrong table, this should be Table 1
Line 1040 (in supp.) vary -> very
Section D.1 in the appendix is empty, maybe D.2 is meant to be a subheading of D.1?
For results tables, it would be good to include a random performance row.

# References
[a] Doughty, Hazel, Dima Damen, and Walterio Mayol-Cuevas. "Who's better? who's best? pairwise deep ranking for skill determination." Proceedings of the IEEE conference on computer vision and pattern recognition. 2018.

[b] Doughty, Hazel, Walterio Mayol-Cuevas, and Dima Damen. "The pros and cons: Rank-aware temporal attention for skill determination in long videos." Proceedings of the IEEE/CVF conference on computer vision and pattern recognition. 2019.

[c] Pan, Jia-Hui, Jibin Gao, and Wei-Shi Zheng. "Adaptive action assessment." IEEE Transactions on Pattern Analysis and Machine Intelligence 44.12 (2021): 8779-8795.

[d] Zhang, Shao-Jie, et al. "Adaptive stage-aware assessment skill transfer for skill determination." IEEE Transactions on Multimedia (2023)

**Questions:**

1. Does the sampling of pairs of videos mean that these might not contain the same action (see above)? Or the actions are sampled first to ensure a wide range of comparison difficulty over actions before video pairs are sampled within action?
2. Were the annotators skilled/experts/knowledgeable in the actions that they were annotating? Or was this found to not be that important given the annotation task? Additionally, how many total annotators were used and were they renumerated for their time?
3. Has the potential bias of the video pairs in the closed task been checked to ensure that naive performance should be 50% instead of video A (or B) occurring as the answer more than 50% of the time. Additionally, I would be interested to know if candidates which could be categorised as C (for insignificant differences) can be understood by the model as this would also increase the naive difficulty to 33% before taking into account a non-uniform distribution.
4. The evaluation protocol for the open-set task seems like it could include errors/inconsistencies depending on the LLM output. Has there been any investigation into this and how much it differs per run and how much it aligns with a human? Currently, the prompts are also given in the appendix with little to no discussion as to why these prompts were chosen, if they were developed over multiple iterations to find the best performing prompt, etc.
5. Did the easy/medium/hard classifications align with experts' opinions for each of the actions? It would be good to know the types of actions that are classed as easy/medium/hard as these are not present within the paper as far as I could tell. It's not clear why an LLM was chosen to do this task.
6. Could more qualitative results and statistics be provided about the dataset? For example, there is very little in the paper regarding the retrieval task of localising the differences: How much of the video does the method need to localise? Are there any temporal biases regarding the timestamps from the videos? Additionally, under the closed set task, more statistics over the number of As, Bs, and Cs that have been annotated and included for each action would be interesting to see. Other statistics that feel like they are missing are the average length of each video (potentially broken down per category) as well as the total number of hours within the dataset.
7. As a thought, has an experiment where the same video is duplicated and provided into the methods, would the output predictions (esp. for the closed set task) give a 50% response rate? Ideally, this is where a method could predict the difference is negligible also.

---

> ### Author Response · Authors · 2024-11-24
> **Comment 1/4 to reviewer dA39**
>
> We’d like to sincerely thank the reviewer for their comprehensive and very thoughtful evaluation. In response, we’ve completed a number of new experiments and human studies, and we’ve made changes to the manuscript. Firstly, we appreciate your recognition of the overall strength of the contributions (rating 4), in particular that the new task is important, and that the benchmark is valuable.
>
> ## References for skill determination
> Thank you for the recommended papers. We have added all four of these to our Related Work. These four works raise the point that video-comparison is a useful signal in model training, even when the supervision is a sparse ranking.
>
> *The remaining text addresses the ‘questions’ section.*
>
> ## Question 1: Are pairs the same action?
> Yes, within each pair, the videos are of the same action. We have adjusted the text to make this more explicit.
>
> ##  Question 2: Importance of annotator expertise
> - The differences in our taxonomy were designed to be straightforward to evaluate for humans. In general, differences were easy to evaluate without additional filtering, likely because each action contained a small number of differences (so they were distinct from each other) and our criteria required all differences to be visually discernible in video. To ensure this was the case, for this review, we re-checked each action to ensure there is no ambiguity, and did fine 3 actions – 2 in surgery and 1 in music – where the actions were arguably a bit ambiguous, and for the final paper we will remove these 3 differences. They only account for 3 of 147 differences.
> - Additionally, to ensure annotation quality, we provided comprehensive instructions with demonstration video pairs for each difference type. As you note, future benchmarks may need to incorporate differences that are even more subtle, and they may require domain expertise. For this benchmark, more discernible actions already lead to a very challenging benchmark.
> - The annotators were college-educated, and remunerated $22.19 per hour.
>
> ## Question 3, part 1 Closed eval – is A/B biased?
> - 49.3% of samples are ‘A’ and 50.7% are ‘B’, so there is no significant dataset bias, and we do not require random swapping at inference time. We’ve added this important detail to Section .3.
> - Additionally, we test the impact of video order on GPT-4o for the `fitness' category, which has samples in the easy and medium subsets(sample size 193). We test flipping the order of videos which flips the A/B answer. The performance is 54.8% in the original evaluation, and reversing the order of videos gives performance of  55.5%, showing a 0.7% difference. This result suggests that the performance on VidDiffBench is not significantly sensitive to video order.
>
> ##  Question 3, part 2, inclusion of Option ‘C’ for Closed Setting
> Thank you for your thoughtful suggestion. Our initial approach to formulating the closed evaluation did include an option ‘C’ for insignificant differences, as you proposed. However, the challenge of calibration made fair evaluation difficult. For example, when comparing two videos of a basketball shot to evaluate stance width, the question arises: how different is “different enough” to be both relevant for skill learning and perceptible? Different annotators may apply varying thresholds for what constitutes a significant difference, leading to inconsistencies. Introducing option ‘C’ further complicates evaluation because it requires calibrating not only the human annotators but also the VLMs, which may have different internal thresholds for perceiving significance. To address these challenges, we adopted the following approach:
> - Annotators were instructed to choose either ‘A’ or ‘B’ only when the difference was clearly perceptible.
> - We limited the evaluation of VLMs to cases where there was a very clear ground truth answer of either ‘A’ or ‘B.’
>
> This method ensures fairness by focusing on scenarios with unambiguous ground truth, avoiding complications introduced by subjective calibration thresholds. While we briefly discuss this in the section on annotation creation, we recognize that this is a nuanced point. Therefore, we have added a more detailed discussion to the appendix for further clarity.

---

> ### Author Response · Authors · 2024-11-24
> **Comment 2/4 to reviewer dA39**
>
> ## Question 4,effectiveness of LLM Evaluation
> Thank you for your question about trustworthiness of LLM evaluation in the open setting. We’ve added these new experiments and human evaluations to the appendix.
> - [Robustness to Multiple Runs] The LLM evaluation is robust to random seed. We repeated the evaluation five times with different random seeds and observed a standard deviation of only 0.7 in the final evaluation score. This indicates that the results are consistent across runs. Although the prompt was specifically engineered for the GPT-4o-2024-08-06 model, we ensured consistency by fixing the model for all evaluations, treating all comparisons under identical conditions.
> - [Comparison with Human Evaluation] To measure alignment with humans, we recruited 3 human annotators to perform open evaluation matching, each with 44 video pairs and 347 individual differences. For each video pair, they were provided with a list of ground truth differences, and asked to match each one to a predicted difference from a list, or to suggest no match. We calculated inter-rater agreement across annotators and the automated LLM system. The results are as below. We can see semantic matching proved to be challenging for humans – the mean of pairwise rater agreement from each human to the other humans was 75.7%. Meanwhile, the mean agreement between our automated system and human annotators was 73.9%. Therefore, our LLM-based approach is on par with human annotators, while being completely automatic.
>
>
> |             | **LLM** | **human 1** | **human 2** | **human 3** |
> |-------------|---------|-------------|-------------|-------------|
> | **LLM**     |         |        72.4 |        74.0 |        70.1 |
> | **human 1** |    72.4 |             |        75.0 |        78.2 |
> | **human 2** |    74.0 |        75.0 |             |        73.9 |
> | **human 3** |    70.1 |        78.2 |        73.9 |             |
> | **avg**     |    72.2 |        75.2 |        74.3 |        74.0 |
>
> - [Details of Prompt for LLM Evaluation] The LLM prompt was carefully developed using a prompt engineering workflow. We selected a set of four evaluation samples, covering two actions and two models, and iteratively refined the prompt based on performance in individual runs. For example, we added the instruction: "Only match entries if their description strings are visually similar, even if the word choices differ." This adjustment was necessary because the LLM struggled to match equivalent descriptions phrased differently (e.g., “the feet stance is wider” vs. “the legs are spread wider apart”). While this approach achieved satisfactory results, we acknowledge that the prompt could be further optimized using more systematic methods, such as DSPy (https://arxiv.org/abs/2310.03714). Exploring such techniques is a promising direction for future work.
>
> ## Question 5, Assigning easy/medium/hard splits by LLMs
> - Choosing the difficulty splits requires a holistic view of all the actions, so we decided it didn’t make sense for experts to suggest them, since they are only familiar with a few actions each. On the other hand, we didn’t want to rank the splits based on performance of current models since this felt like biasing towards current models; and besides, the performance for many actions in ‘medium’ and ‘hard’ is already random, so it would be hard to differentiate these actions. LLMs are a good candidate because they have a good understanding of the actions and are relatively free of the biases of this paper’s authors Furthermore, human annotators could not do the ranking, because no human annotated all the actions.
> - To further support the choice of an LLM,  we asked 3 humans to rank the action comparisons from easiest to hardest, and compared against the LLM ranking. We then computed the Spearman’s rank correlation between all ranking sets, and the results are in the below table. The mean of the pairwise correlations between the humans was 0.602, while the mean of pairwise correlations between the LLM and humans was higher at 0.673. This shows (i) that there is non-negligible variability in human rankings, and (ii) that the LLM ranking is reasonable, and actually better correlated with most humans compared to several of the human annotations.
>
>
> |             | **LLM** | **human 1** | **human 2** | **human 3** |
> |-------------|---------|-------------|-------------|-------------|
> | **LLM**     |         |       0.531 |       0.680 |       0.806 |
> | **human 1** |   0.531 |             |       0.459 |       0.645 |
> | **human 2** |   0.680 |       0.459 |             |       0.703 |
> | **human 3** |   0.806 |       0.645 |       0.703 |             |
> | **avg**     |   0.673 |       0.545 |       0.614 |       0.718 |
> - We added a new appendix table showing the difficulty splits with lists of actions and full descriptions.

---

> > ### Author Response · Authors · 2024-11-24
> > **Comment 3/4 to reviewer dA39**
> >
> > ##  Question 6, Dataset attributes Thank you for these suggestions. We’ve included more detailed dataset statistics. In the table below we show these statistics broken down by difficulty split. In the appendix we additionally show these broken down by action.
> >
> > | Split   | # video pairs | Avg video length (secs) | Total video length (mins) | # differences tagged | StdDev within retrieval type | StdDev across retrieval types | Difference annotations count | Difference annotations A/B/C distribution |
> > |---------|---------------|-------------------------|---------------------------|----------------------|------------------------------|-------------------------------|------------------------------|-------------------------------------------|
> > | easy    |            95 |                     2.1 |                       6.5 |                 1224 |                         8.4% |                         17.3% |                          578 | 167/190/221                               |
> > | medium  |           265 |                     3.9 |                      34.7 |                 4788 |                         5.2% |                         25.7% |                         1771 | 622/605/1143                              |
> > | hard    |           197 |                    18.7 |                     122.5 |                 3542 |                         4.1% |                         20.2% |                         2370 | 435/452/884                               |
> > | Overall |           557 |                     8.8 |                     163.7 |                 9554 |                         5.9% |                         21.0% |                         4719 | 1224/1247/2248                            |
> >
> > - Average video length is longer as the difficulty gets higher: 2.1/3.9/18.7 seconds, for easy/medium/hard. Compared to video QA datasets, the lengths are relatively shorter because we focus on fine-grained action understanding in how actions are performed. The total length of videos is 163 minutes.
> > - [Retrieval tags, temporal bias] For the ‘retrieval tags’, we first show the number of retrieval tags – 9554 total. To give insight into their distribution within each video, each instance is normalized to the video length, and compute its ‘video location’. E.g. in a squat, the starting position might be position 0.1, the bottom of the descent 0.45, and the squat finish at 0.87. Within each retrieval type, we compute “StdDev within retrieval type”, which intuitively measures how well-aligned are the key points in the video. For example, if the average squat video records ‘bottom of descent’ at location 0.45, and “within StdDev” is 0.06, then the mean distance from the average is 0.06 (so at 0.39 or 0.51). The “within StdDev” is on average 0.059, indicating there is some variation in retrieval position, but there is temporal bias. This is expected since each video is trimmed and contains an atomic action. Future benchmarks could use untrimmed videos to make retrieval annotations less aligned, but the present benchmark is already difficult for SOTA models, so this is unnecessary now.
> > - [Retrieval tags, coverage] We also measure ‘StdDev across retrieval types’, meaning the standard deviation of different retrieval classes within one video. Intuitively this measures how much of the video is ‘covered’ by retrieval keypoints. This is 0.21 on average. So if the mean of retrieval keypoints were 0.5, then the average retrieval annotations is around 0.29 or 0.71 in the video.
> > - Additionally, we have shown the count of difference annotations and the A/B/C distribution; the ‘no difference’ annotation of ‘C’ is the most prevalent.
> >
> > ### Question 7, Experiment with duplicating video
> > The idea of passing an identical video as A and B to the system is interesting. As suggested, we tried it on the closed setting, and applied all ‘easy’ subset for GPT-4o. Over two random seeds, the results were 49.3 and 50.2. This is an interesting validation check that the benchmark passes. We also added this to the Appendix.

---

> > > ### Author Response · Authors · 2024-11-24
> > > **Comment 4/4 to reviewer dA39**
> > >
> > > ### Clarity and Paper Structure
> > > Thank you for your feedback. We appreciate your recognition of the sentence-level grammar and clarity in our writing. However, we acknowledge your concerns regarding the structure of the paper and its impact on readability. Based on your suggestions, we have made several improvements to enhance the paper's organization and flow:
> > > - Earlier Task Definition: We have moved the task definition earlier in the paper, ensuring that readers have a clear understanding of the new task before encountering details of the dataset.
> > > - Dedicated Method Section: The VidDiff staged method now has its own dedicated section, separate from the experimental results. This change provides a clearer and more focused explanation of the method, aligning with its positioning as a key contribution in the introduction.
> > > - Revised Benchmark Section: To address the issue of key information being split between the main text and the appendix, we have revised the benchmark section to ensure that all essential details are included in the main text. References to the appendix are limited to the start or end of relevant sections, directing readers to supplementary information without requiring frequent flipping between sections.
> > > These revisions aim to make the paper easier to read and follow, particularly for readers unfamiliar with the new task. We hope these changes address your concerns and improve the overall readability and structure of the manuscript.

---

> ### Comment · Reviewer_dA39 · 2024-11-25
>
> Thank you for answering my questions and providing more comments about the paper. Because of this, I will increase my rating towards this paper. I believe the extra experiments that were conducted for the validation towards the benchmark to be interesting and informative. I do not agree with the reasoning behind the LLM for the easy/medium/hard split, but do not believe that to be large enough to sway my overall decision of the paper. However, upon reading the updated version of the paper, I discovered some minor spelling/grammatical errors:
>
> Line 141: "... the supervision signal is commonly a binary of which..." -> either "commonly binary", or "commonly a binary one"
>
> Line 144: "... comparison while having a large scale." -> "whilst being large scale."
>
> Line 311: "... the A/B/C ration is ..." -> "... the A/B/C/ ratio is ..."
>
> Line 249: "... key challenge of aligning precise temporal alignment..." align[ing/ment] is repeated here, can aligning be dropped?

---

> > ### Author Response · Authors · 2024-11-26
> >
> > Thanks for supporting the work by raising the score! And we appreciate you reading the paper closely and identifying the issues, which are now fixed in the revised pdf.

---

### Official Review · Reviewer_ThzX · 2024-11-01

**Soundness:** 3
**Presentation:** 2
**Contribution:** 3
**Rating:** 6
**Confidence:** 4

**Summary:**

The authors introduce a method and dataset designed to compare subtle action differences across video pairs. Their method, VidDiff, uses a three-stage process to generate, localize, and verify these differences using multimodal models.

Main Contributions:

- A Dataset includes 557 video pairs across domains like sports, fitness, and surgery, annotated with over 4,700 fine-grained action differences. The dataset is designed to help models learn and evaluate nuanced action comparisons.
- A Framework uses a three-step pipeline to identify differences: (1) generating potential differences with a language model, (2) aligning frames between videos using CLIP and Viterbi algorithms, and (3) verifying differences with a vision-language model.
- A method is compared with leading multimodal models (e.g., GPT-4o and Gemini-1.5 Pro), showing improved performance in closed-set settings. This comparison also highlights challenges in current models for frame alignment and action detail recognition.

**Strengths:**

- The method is shown to outperform baseline large multimodal models by systematically isolating action differences through a three-stage approach, excelling in both closed and open settings.
- The introduction of benchmark, with extensive annotations across varied domains (e.g., fitness, surgery, sports), provides a unique and structured dataset for fine-grained video action comparison.
- Evaluations and ablation studies demonstrate the robustness and effectiveness of the method, especially in tasks that require precise frame alignment and action differentiation.
- The proposed task and methods address real-world challenges in skill-based learning environments.

**Weaknesses:**

- The performance of the leading multimodal models on the dataset is not clearly demonstrated, and examples comparing success and failure cases across models would enhance understanding of their effectiveness.
- The Difference Proposer stage’s reliance on large language models (LLMs) may introduce biases or inaccuracies, especially when generating action differences for complex or nuanced tasks. Providing more details on the generated proposer queries and their corresponding ground truth labels would enhance clarity.
- Although the multi-stage approach is well-structured, it presents a risk of error propagation. Inaccuracies in early stages could impact the final outputs, potentially reducing overall reliability, particularly in the open-set task, where the Difference Proposer’s effectiveness is not fully evaluated.
- While the paper introduces a detailed taxonomy for annotations, the reasonableness of some annotations remains unclear. For example, the “surgery_0” annotation includes the description "the movements in video A are more efficient than in video B," which lacks a concrete definition and could be interpreted inconsistently even by human evaluators. Scaling this annotation approach to larger datasets or adapting it to new domains could also present significant challenges.
- Minor issue: The table mentioned as Table 7 in Section 5.3 is missing.

**Questions:**

Please refer to weakness for more details.

---

> ### Author Response · Authors · 2024-11-24
> **Comment 1/2 for reviewer ThzX**
>
> We’d like to thank reviewer ThzX for the comprehensive and constructive comments, and for highlighting that the task is well-motivated, that the benchmark is well-constructed, and that the method’s results are interesting. We now address each of the given concerns:
>
> ### Comparing SOTA models
> We have performed a more thorough comparison of the different SOTA LMMs on VidDIffBench, added a small subsection to the results, and a discussion in appendix. Specifically we look at each action, and compare the different LMMs.
> - First, we show the correlations in the per-action scores between models, which is interesting:
>
> |                   | GPT   | Gemini | Claude    | LLava-Video | Qwen2-VL  |
> |-------------------|-------|--------|-----------|-------------|-----------|
> | GPT-4o            |       |  0.152 | **0.375** |       0.243 |     0.273 |
> | Gemini-1.5-Pro    | 0.152 |        |     0.215 |       0.111 |     0.223 |
> | Claude-3.5-Sonnet | 0.375 |  0.215 |           |       0.261 |     0.220 |
> | LLaVA-Video       | 0.243 |  0.111 |     0.261 |             | **0.376** |
> | Qwen2-VL-7b       | 0.273 |  0.223 |     0.220 |       0.376 |           |
>
> - The correlations are generally low, but there are 3 clusters of models. LLaVA-Video and Qwen2-VL are in a cluster; they are both open-source, and have the same LLM backbone. Then GPT-4o and Claude-Sonnet cluster together, and Gemini is not similar to any other model. We can speculate that for video data, Claude and GPT have similar training strategies, while Gemini’s is different.
> - Next we compare model performance within one action, and this is over two large tables in the Appendix. Specifically we measure ‘relative performance’: the difference between the model score on that action compared to the mean score across all models for the action. The most significant results in the benchmark are on the easy split. Here, the improvement in score is uniform for all models The models are generally close to each other. The ‘relative performance’ is usually less than 10 points – when it is higher, the sample size is very small.
> - By comparing models at the level of actions, we are considering smaller sample sizes than in the main results, which compare models at the level of easy/medium/hard splits. There is therefore lower statistical power to identify significant result differences, so the results are less certain. We elected not to compare model performance at the level of action differences, because here the sample sizes are very small, so any correlations would not meet significance thresholds.
>
> ### LLMs as Difference Proposers
> In our work, we introduced two types of evaluation: a closed-set setting and an open-set setting. The closed-set setting relies on human-annotated difference proposals, eliminating the need for LLMs. This setting simulates scenarios where specific differences of interest are already known and serves to evaluate the VLM’s video comparison capabilities directly. In contrast, the open-set setting leverages an LLM-based difference proposer to generate action differences, aiming to closely mimic real-world, end-to-end conditions. This approach not only tests the model’s ability to identify differences in video content but also evaluates its language reasoning capabilities. By incorporating both settings, our framework addresses varying levels of task complexity and grounds the evaluation in both controlled and practical contexts.
> - To assess the LLM proposer’s performance, we compare its generated differences with human-annotated differences, reporting matching accuracies across three levels of task difficulty. These results demonstrate that the LLM-based proposer can generate accurate difference proposals more than 60% of the time, making it a viable and useful component in our open-set evaluation.
> - Easy: 68.9%
> - Medium: 61.9%
> - Hard: 62.0%
>
>
> CONTINUED .....

---

> ### Author Response · Authors · 2024-11-24
> **Comment 2/2 for reviewer ThzX**
>
> ### Error Propagation in Multi-Stage Method
> - This is a good question, so thank you for your comment.
> - The multi-stage compound system and the single-stage end-to-end system each have their respective strengths and limitations. While the multi-stage approach indeed carries the risk of error propagation, single-stage systems face challenges in performing complex multimodal reasoning effectively. As shown in Tables 2 and 3 in our manuscript, our method demonstrates superior performance compared to single-stage systems (VLMs), highlighting the potential advantages of the multi-stage approach in addressing this task.
> - In this work, our primary contribution lies in defining a new task and providing a benchmark to explore it. The multi-stage system presented serves as a baseline to illustrate the potential of this approach. We believe that more advanced single-stage VLMs, particularly those equipped with enhanced internal reasoning capabilities (e.g. GPT4-o1), could further improve performance in this domain.
> - Additionally, we have evaluated the performance of the LLM-based Difference Proposer, as detailed in the above provided tables, to address its effectiveness and demonstrate its role in the overall system.
>
> ###  Annotation taxonomy
> - Thank you for your constructive comment. Firstly, note that the majority of the candidate differences are clearly visually discernible, and we argue they can be evaluated objectively. You are correct that there are a small number of differences that do require more interpretation. To address this, we do the following
> - We performed a manual review of all differences and found 3 cases that are potentially ambiguous –two in surgery, and 1 in music. This is therefore a potential issue for only 3 of the 147 differences.
> - Note that we included these actions because we were weighing the objectivity of the difference vs the importance of the difference. Our expert-informed taxonomy generation process identified these as important differences. Furthermore, for surgery and music, the same person did the annotations within a single difference. The annotation instructions emphasized to only mark “A” or “B” if the magnitude of difference is clear, and this was often the case because the datasets have a wide range of skill levels. So we argue that the risk of inconsistency is minimal.
> - Having said that, we agree that these differences do add concerns about the objectivity of labels. Since it makes up such a small part of this dataset, we have decided to remove them  – we will update the results in the final manuscript, which will impact only the medium and hard results to a small degree (the LMM performance for those actions was already random).
>
> ### Table number
> Thank you, we have corrected to reflect that this is an Appendix table, since it is so long.

---

> > ### Comment · Reviewer_ThzX · 2024-11-25
> >
> > Thank you for your response. It addresses most of my concerns, and I will adjust my evaluation to give this paper a higher score.

---

> > > ### Author Response · Authors · 2024-11-26
> > >
> > > Thank you for reconsidering and raising your score; we appreciate your thoughtful feedback and support for publication

---

### Official Review · Reviewer_nLbM · 2024-11-02

**Soundness:** 3
**Presentation:** 4
**Contribution:** 3
**Rating:** 6
**Confidence:** 4

**Summary:**

This paper introduces Video Action Differencing, a novel task of identifying subtle differences between videos of the same action. It also introduces a new benchmark sourced from mutliple video datasets with new annatations. A new method is proposed for this new task with state-of-the-art performance.

**Strengths:**

1. The proposed task has not been explored, which has key applications for some scenarios in real life.
2. The construction process for the dataset is technically sound with different splits.
3. The visulizations are clear and interesting.

**Weaknesses:**

Weakness and questions:
1. Do the authors consider factors like fps for videos, which may impact the restults of answering questions like "the speed of the arms is faster" for distinguishing videos A and B.
2. For the open-set benchmark, have the authors analyzed the reasons for why QWen2-VL performs so worse?
3. Have the authors visualized the selected frames by the frame localizer compared to the ground-truth frames? What's about the effects of frame localizer compared to the ground-truth frames?

**Questions:**

see above

---

> ### Author Response · Authors · 2024-11-24
> **Comment 1/2 for reviewer nLbM**
>
> We’d like to thank reviewer nLbM for recognising the strength of the work, especially that the proposed Video Action Differencing task is novel and well-motivated, that the benchmark construction process is sound, and that visualizations add understanding to the results.  We now hope to address lingering issues:
>
>
> ### Impact of fps
>
> We strongly agree that fps is an important consideration for evaluating fine-grained actions.
> - While typical video benchmarks like Video-MME sample videos at 1fps, we have sampled at a higher rate depending on category. The categories with shorter videos were sampled at a higher rate: 4fps for ‘fitness’, 5fps for ‘ballsports’, and 6fps for ‘diving’ (they are different so they can be compatible with fps in the source dataset). We chose this relatively higher rate because we are interested in more fine-grained differences, though we did not sample higher due to practical cost constraints of processing too many frames. The longer videos ‘surgery’ and ‘music’ were sampled at 1fps: these are longer videos where differences are discernible at lower sampling rates, and where the longer videos make high-fps sampling impractical.
> - To show that our fps is reasonable, we tested the three closed-source models on a range of fps levels on the ‘easy’ subset of closed evaluation. We chose this set because this is where statistically significant differences were clear. The results are in this table:
>
> |                   | 1fps | 2fps | 4fps | 8fps | avg  |
> |-------------------|------|------|------|------|------|
> | GPT-4o            | 58.0 | 59.4 | 58.8 | 59.1 | 58.8 |
> | Gemini-1.5-Pro    | 59.7 | 66.9 | 65.8 | 66.9 | 64.8 |
> | Claude-3.5-Sonnet | 58.1 | 58.5 | 56.6 | 52.9 | 56.5 |
>
> - **Across all models, the sampling rate that we use, 4 fps, has reasonable scores**. For all models, the variability is low: GPT’s scores are within 0.8 points of the average; all other models have scores within 2.1 points of the average (except for the low sampling rate of 1fps in Gemini, where it degrades by 5.2 points). Moreover, the optimal fps is different for different models.
> - To help explain the results, we refer to the qualitative examples in the main results sections. The only ‘success cases’ for all our models were those having easy localization, and coarse differences. We hypothesize that fps is not important for these cases. Where fps is likely important – fine-grained multiframe reasoning – the current LMMs cannot perform better than random. So although 2fps currently has good performance, we believe that as LMMs improve, they will perform better on subtle motions and using a higher fps will be important.
> - We have added these points to the main document, and added the detailed results in the Appendix.
>
> ### Qwen2-VL poor open performance
> This is a good suggestion, and we’ve performed a deeper analysis into the lower scores of Qwen2-VL-7B. We found that a key issue here is that **Qwen2-VL-7b was failing to follow the evaluation prompt**, while the other compared models did follow it. Below are more evaluation details:
> - We sampled 3 video pairs for each action and manually inspected Qwen’s responses, identifying multiple key issues. Below, we list each issue, and provide a quantitative estimate for the prevalence of each issue.
>   - (45% of differences) Proposing differences not relevant to *how* to perform actions, but instead are visual things like “The person in video a is wearing a blue jacket, while the person in video b is wearing a plaid shirt.” We estimated prevalence by using a gpt-4o query that we manually prompt engineered.
>   - (26% of differences) Proposing a difference that is actually not a difference, e.g. “The person in video a is performing the exercise with their arms out to the sides, while the person in video b is performing the exercise with their arms out to the sides.” We estimated prevalence by using a gpt-4o query that we manually prompt engineered.
>   - (56% of differences) are repeated, meaning when trying to propose multiple differences, it proposes the same difference multiple times.  We could directly measure this prevalence exactly.
>   - (23% of actions) Proposing only a small number of differences – less then half as many as what is prompted for. We could directly measure this prevalence exactly.
>   - (<5% of differences) Proposing vague differences that are harder to interpret visually like “The player in video a has a more versatile and adaptable skill set than the player in video b”. We estimated prevalence by using a gpt-4o query that we manually prompt engineered.
> - Overall, only 31.9% of proposed differences by Qwen did not suffer from any of these errors. (Note that some differences suffered from multiple errors at the same time)
>
> CONTINUED .....

---

> ### Author Response · Authors · 2024-11-24
> **Comment 2/2 for reviewer nLbM**
>
> .... CONTINUING FROM LAST COMMENT
>
> ### Frame localization
> - We did ablate the importance of the frame localizer in the main Table 5, in which we fix the frame-level action differencing module (the 3rd stage) and only implement the frame localizer module with different methods.
> -  for the easy split. This shows that choosing a random frame to localize leads to accuracy of 50.1%, our approach gives 65.8%, and using the ground truth frames gives 78.6%.
>
> |              Ablation             | Accuracy (closed, easy split) |
> |---------------------------|-------------------------------|
> | Oracle (GT timestamps)    | 78.6                          |
> | Random                    | 50.1                          |
> | Ours w/o Viterbi Decoding | 57.4                          |
> | Ours                      | 65.8                          |
>
> - We visualize some of the selected frames by the frame localizer compared to the ground-truth frames, and found most of these frames are close to GT frames, we add this visualization to the appendix.

---

> > ### Comment · Reviewer_nLbM · 2024-11-25
> > **Response**
> >
> > Thanks for the response. It addresses most of my concerns. And i will keep my score.

---

> > > ### Author Response · Authors · 2024-11-26
> > >
> > > Thanks again for your input, and for supporting acceptance !

---

### Author Response · Authors · 2024-11-24
**Summary of all reviewer responses (1/2)**

We thank the reviewers for their valuable feedback and for recognizing the novelty of our work as well as the contributions of our proposed task and benchmark. The reviewers provided many constructive suggestions, which we have carefully addressed. In response, we conducted several new experiments and revised both the main paper and supplementary materials. All text changes are highlighted in blue for clarity. We believe we have thoroughly addressed each point raised in the reviews and significantly improved the manuscript as a result.

## Summary of most important new results
We now briefly summarize the most important new results

### More benchmarking
We added more models to our evaluation baselines: Claude-3.5-Sonnet, and the recently released LLaVA-Video-7B. Here are the updated results for closed evaluation, showing that LLaVA-Video exceeds Qwen2VL-7B, while Claude outperforms open-source models, but not GPT and Gemini.

|                       | Easy  | Med   | Hard  | Avg   |
|-----------------------|-------|-------|-------|-------|
| **GPT-4o**            | 58.8% | 53.0% | 50.1% | 54.0% |
| **Gemini-1.5-Pro**    | 65.8% | 51.9% | 49.8% | 55.8% |
| **Claude-3.5-Sonnet** | 56.6% | 53.5% | 48.3% | 52.8% |
| **LLaVA-Video**       | 56.6% | 52.0% | 48.3% | 52.3% |
| **Qwen2VL-7B**        | 49.0% | 52.6% | 49.6% | 50.4% |
| **VidDiff (ours)**    | 65.3% | 55.4% | 50.4% | 57.0% |

### Verification of open eval
Our open evaluation setting requires matching ground-truth difference strings to predicted difference strings, and we propose to use an LLM for that purpose. We added two experiments to verify the trustworthiness of this approach.

First, it is robust to random seed: over 5 runs, the standard-deviation was 0.7 points.

Second, we recruited 3 human annotators to perform matching, and then we computed inter-annotator agreement scores:

|             | LLM  | human 1 | human 2 | human 3 |
|-------------|------|---------|---------|---------|
| **LLM**     |      |    72.4 |    74.0 |    70.1 |
| **human 1** | 72.4 |         |    75.0 |    78.2 |
| **human 2** | 74.0 |    75.0 |         |    73.9 |
| **human 3** | 70.1 |    78.2 |    73.9 |         |
| **avg**     | 72.2 |    75.2 |    74.3 |    74.0 |

This shows that (i) the matching task is challenging, with mean agreement amongst humans at 75.7%, and (ii) LLMs have comparable agreement to the humans at 72.2%.

This supports that LLM-based matching is reasonable, allowing us to enjoy the benefits of automatic evaluation that is consistent and reproducible.

### LLMs in assigning splits
We chose LLMs for determining easy/medium/hard difficulty splits because they have action understanding, and because this avoids biasing towards either current models or towards the opinions of authors. We recruited 3 humans to rank the actions by difficulty, and computed the Spearman rank correlation between the LLM and all humans:

|             | LLM   | human 1 | human 2 | human 3 |
|-------------|-------|---------|---------|---------|
| **LLM**     |       |   0.531 |   0.680 |   0.806 |
| **human 1** | 0.531 |         |   0.459 |   0.645 |
| **human 2** | 0.680 |   0.459 |         |   0.703 |
| **human 3** | 0.806 |   0.645 |   0.703 |         |
| **avg**     | 0.673 |   0.545 |   0.614 |   0.718 |


The mean of the pairwise correlations between the humans was 0.602, while the mean of pairwise correlations between the LLM and humans was higher at 0.673. This shows (i) that there is non-negligible variability in human rankings, and (ii) that the LLM ranking is reasonable, and actually better correlated with most humans compared to several of the human annotations.

### Baseline model comparison
The main results table compares the different models at different splits, but for a more fine-grained comparison, we computed the accuracy on a per-action basis. Given these scores, we then computed the correlations between the models:


|                       | GPT   | Gemini | Claude | LLava-Video | Qwen2-VL |
|-----------------------|-------|--------|--------|-------------|----------|
| **GPT-4o**            |       |  0.152 |  0.375 |       0.243 |    0.273 |
| **Gemini-1.5-Pro**    | 0.152 |        |  0.215 |       0.111 |    0.223 |
| **Claude-3.5-Sonnet** | 0.375 |  0.215 |        |       0.261 |    0.220 |
| **LLaVA-Video**       | 0.243 |  0.111 |  0.261 |             |    0.376 |
| **Qwen2-VL-7b**       | 0.273 |  0.223 |  0.220 |       0.376 |          |

The correlations are generally low, but there are 3 clusters of models. LLaVA-Video and Qwen2-VL are in a cluster; they are both open-source, and have the same LLM backbone. Then GPT-4o and Claude-Sonnet cluster together, and Gemini is not similar to any other model. We can speculate that for video data, Claude and GPT have similar training strategies, while Gemini’s is different.

---

> ### Author Response · Authors · 2024-11-24
> **Summary of all reviewer responses (2/2)**
>
> ## Other experiments
> We completed a number of other experiments and paper updates:
> - An ablation over frames sampling rate (fps) on baseline LMMs, showing that our choices were reasonable (reviewer nLbM)
> Further datasets statistics, especially giving insight into video lengths and temporal biases in the retrieval / localization task (reviewer dA39)
> - Significant changes to the structure of the paper for improved clarity (reviewer dA39), and smaller changes for clarity (reviewer 8a8a and dA39).
> - Longer discussion of related video-pair datasets (reviewer 8a8a), and added new related works on paired datasets (reviewer dA39).
> - Analysis into poor results for QwenVL-2 on open evaluation, concluding that it struggled with text instruction following. (nLbM )
> - A section on the effectiveness of the Proposer module in our multistage VidDiff (ThzX)
> - For the benchmark, we filtered 3 out of the 147 differences to give high confidence that all differences are visually discernible objectively (reviewer dA39 and ThzX)
> - Showed that there is no potential bias due to the ordering of A/B in the multiple questions  (reviewer dA39) by performing experiments with flipping video order.
> - We further justified our design decisions in the closed evaluation setup (dA39), in the multistage design (reviewer ThzX), and in the annotator instructions (reviewer 8a8a)

---

### Author Response · Authors · 2024-12-03
**Author summary of the discussion period.**

Thanks again to all the reviewers for engaging thoughtfully during the discussion period. The comments raised many interesting points, and have greatly improved the paper. The main paper and supplementary pdfs have been updated, with changes marked in blue.

Our paper introduces Video Action Differencing with three key contributions: a novel task, a benchmark (VidDiffBench), and a multi-stage method (VidDiff).

During the rebuttal period, we added new experiments and improved the writing to address the reviewer’s concerns. The most significant were:
- Improved breadth of benchmarking by adding new large multimodal models (LMMs), specifically Claude-3.5-Sonnet and LLaVA-Video.
- Validated the use of LLMs in the evaluation protocol for the ‘ open-set' setting, using human studies and robustness tests.
- On dataset quality, showed that there is negligible label bias, removed a small set of potentially ambiguous action differences, and added text to the manuscript describing dataset statistics and comparisons to prior datasets.
- Better justified the motivation for the task, especially arguing that language is a natural way to receive feedback in skill learning.
- Supported the benchmark split into easy/medium/hard using a human study.
- Added ablation studies over frame sampling rates (fps) and showed the robustness of our design choices across multiple LMMs.
- Deeper analysis into the failure cases of QwenVL-2 in open evaluation, finding issues with instruction-following.
- Restructured the paper's organization and expanded related work sections to better position our contributions in the context of video-pair datasets

As a result of these discussions, all reviewers concluded the open discussion period with scores above the acceptance threshold.

---

### Meta-Review · Area_Chair_w9ZD · 2024-12-20

**Metareview:**

The paper introduces a new task and a benchmark for action differencing, to tell the differences between different actors performing the same action. Various baselines (included latest LLM-based baselines) are compared with the proposed approach, VidDiff.

All reviewers appreciate the novelty of the task and recommend to accept the paper. Most of the discussions are about further clarifications and asking additional ablations and analysis. The author(s) did a great job in providing additional experiment as well as further elaborate on motivation and insights which help to convince all reviewers to support accepting the paper. The area chair reads all reviews and discussions and agrees with the reviewers, thus recommends an acceptance.

**Additional Comments On Reviewer Discussion:**

Most of the discussions are about further clarifications and asking additional ablations and analysis. The author(s) did a great job in providing additional experiment as well as further elaborate on motivation and insights which in results turn all reviewers to support to accept the paper.

The area chair reads all reviews and discussions and agrees with the reviewers, thus recommends an acceptance.

---

### Decision · Program_Chairs · 2025-01-22

Accept (Poster)